# Digital automation of transdermal drug delivery with high spatiotemporal resolution

Yihang Wang[1,6], Zeka Chen[2,6], Brayden Davis[3], Will Lipman[4], Sicheng Xing [3], Lin Zhang[1], Tian Wang[3], Priyash Hafiz[3], Wanrong Xie[1], Zijie Yan[1], Zhili Huang [5], Juan Song [2] ✉ & Wubin Bai [1] ✉

Transdermal drug delivery is of vital importance for medical treatments. However, user adherence to long-term repetitive drug delivery poses a grand challenge. Furthermore, the dynamic and unpredictable disease progression demands a pharmaceutical treatment that can be actively controlled in real-time to ensure medical precision and personalization. Here, we report a spatiotemporal on-demand patch (SOP) that integrates drug-loaded microneedles with biocompatible metallic membranes to enable electrically triggered active control of drug release. Precise control of drug release to targeted locations (<1 mm$^2$), rapid drug release response to electrical triggers (<30 s), and multi-modal operation involving both drug release and electrical stimulation highlight the novelty. Solution-based fabrication ensures high customizability and scalability to tailor the SOP for various pharmaceutical needs. The wireless-powered and digital-controlled SOP demonstrates great promise in achieving full automation of drug delivery, improving user adherence while ensuring medical precision. Based on these characteristics, we utilized SOPs in sleep studies. We revealed that programmed release of exogenous melatonin from SOPs improve sleep of mice, indicating potential values for basic research and clinical treatments.

Skin provides convenient access for delivering most biotherapeutics and vaccines, typically via a hypodermic needle. However, ensuring patient compliance with long-term, repetitive pharmaceutical treatments remains a significant challenge. Hypodermic injection provides a low-cost and rapid method of drug delivery, but concerns about safe disposal, potential transmission of bloodborne pathogens, and the need for trained personnel hinder its widespread implementation. Recently, advancements in transdermal drug delivery strategies, such as sonophoresis[1,2], iontophoresis[3,4], electroporation[5,6], photomechanical waves[7,8], heat[9,10], microneedles (MN), and others[11], have improved drug permeation through skin, providing safe and painless operations[12] with easy-to-use procedures[13,14]. However, the lack of an automated mechanism for active, precise, and coordinated drug administration over extended periods hinders their applicability for chronic pharmaceutical management. This problem becomes particularly critical for chronic diseases[15,16], including diabetes[17], hyperlipidemia[18], asthma[19], depression[20], and others, where repetitive drug administrations are required, and a dynamically personalized delivery schedule could improve drug efficacy and decrease drug toxicity[21]. However, most existing drug delivery devices have limited capability to automate delivery digitally, especially outside hospital settings and in a comfortable, long-lasting fashion[22].

[1]Department of Applied Physical Sciences, University of North Carolina at Chapel Hill, Chapel Hill, NC 27599, USA. [2]Department of Pharmacology, University of North Carolina at Chapel Hill, Chapel Hill, NC 27599, USA. [3]UNC/NCSU Joint Department of Biomedical Engineering, Chapel Hill, NC 27599, USA. [4]Department of Psychology and Neuroscience, University of North Carolina at chapel Hill, Chapel Hill, NC 27599, USA. [5]State Key Laboratory of Medical Neurobiology, Fudan University, Shanghai 200032, China. [6]These authors contributed equally: Yihang Wang, Zeka Chen. ✉e-mail: juansong@email.unc.edu; wbai@unc.edu

Microneedles have shown great promise in facilitating the delivery of various types of drugs, including small molecules[23,24], peptides[25], nucleic acids[26,27], and nano composites[28–33]. Furthermore, modulating the structural integration and chemical functionalization of MNs enables a broad range of release profiles. For example, MNs with a core-shell structure that hosts a drug reservoir inside each needle can exhibit a pre-programmed, multi-step release profile with tunability via designing the degradation time of MN shell layers[34]. This method also allows the integration of multiple drugs to enable sequential release as a combined therapy[35]. However, the complexity involved in fabrication poses challenges for manufacturing scalability, and once deployed, it becomes difficult to modify the pre-programmed release time of the microneedles. Chemical functionalization on MNs provides a solution to introduce self-sensing and self-responsiveness capabilities. For insulin delivery, researchers use specific chemical groups to functionalize the material of MNs, such as phenylboronic acid or aminoimidazole[25,36], which react with glucose in body biofluids and induce structural changes in the polymer network of MNs[32]. This triggers the release of embedded drugs in response to glucose levels in the surrounding environment, enabling convenient and self-regulated long-term drug release to control chronic diseases. However, the complexity of synthesis and the reliance on local microenvironments rather than global body physiology limit its practical applicability.

In addition to passive drug release, the potential of active control to deliver drugs on demand through external triggers has paved the way for closed-loop therapeutic systems when integrated with associated health monitors, enhancing treatment precision and dynamics. The heat-triggered release represents a typical example where MNs with drug loaded inside a thermally responsive material (e.g., Expancel microspheres and paraffin $C_{23}$[37,38], which exhibit significant expansion in pore or volumetric size upon heating at, typically, 47 °C, to secrete the drug accordingly. Mechanisms such as these allow drug release to be controlled with thermal triggers via an electrical heater or optical illumination[39,40]. However, the inability to effectively focus thermal energy in a confined space of biological tissue precludes its delivery precision and safety. Thin membranes based on biocompatible metals serving as a gate for drug reservoirs can enable actively controlled drug delivery[41,42]. The metallic gates can disintegrate or dissolve upon an electrical trigger[43]. Demonstrated devices exploit various biocompatible metals, including Mg (-30 µm), Mo (-10 µm), and Au (-300 nm), as the metal gates to form electronic implants that enable on-demand drug delivery[44,45]. Opening of the metallic gates by either anodic oxidation (typically for Mg and Mo) or corrosion-induced crevices (typically for gold) via electrical triggers can effectively initiate the active release behavior. The compatible integration via microfabrication technologies enables such drug-releasing mechanisms with high spatiotemporal controllability via small electrical signals (typically, +1.04 V vs. SCE)[43,46], which could potentially realize pharmaceutical automation in space and time.

Here, we present a skin-interfaced drug delivery system that utilizes electrically triggered, gated MNs to realize on-demand drug delivery with high spatiotemporal controllability. The drug delivery system, named spatiotemporal on-demand patch (SOP), uses a thin gold layer (-150 nm) coated onto MNs to enable drug encapsulation and protection at the standby stage. Small electrical triggers (-2.5 V, DC) for 30 s effectively disintegrate the gold coating, which exposes the drug to initiate delivery. Microfabrication processes enable circuitry designs of the gold layer to realize release triggering of individual MNs or subsections through a wireless communication module (e.g., Near-field communication, Bluetooth Low Energy). Direct deposition of the gold layer onto MNs overcomes limitations in the fabrication complexity and device robustness associated with the reservoir systems with free-standing metallic gates, as reported previously in implantable devices[43]. Ultrafine spatial control (<1 mm²) of drug release of single MN, active management with high temporal (less than 30 s) resolution of drug release, wireless operation, and comfort wearability highlight the enabling capabilities of the SOP. Both benchtop experiments using a fluorescent dye and in vivo study through intracranial injection demonstrate the high potential of SOP as a general, fully wireless, wearable platform for personalized, chronic drug delivery to improve pharmaceutical efficacy and user adherence. Moreover, in vivo demonstration via intracranial injection of SOP reveals its potential utility for neural therapy and modulation. The high spatiotemporal resolution and SOP's on-demand drug release feature make it an advanced tool for brain research with model animals, especially in studying neural circuits mapping and cause-and-effect relationships between neural activity and behavior[47]. Capabilities offered by SOP to deliver therapeutic agents sequentially and proactively to specific brain regions associated with neurological disorders may deepen our understanding of the underlying mechanisms of their pathologies. This insight can lead to the development of more targeted treatments for disorders like Parkinson's disease, epilepsy, depression, and Alzheimer's disease[48–52].

## Results and discussion

### Configuration and characterization of the spatiotemporal on-demand patch (SOP)

Figure 1a highlights the design of an SOP, which consists of two primary parts: i) a drug-loaded microneedle (MN) patch protected with an electrochemically triggerable metal layer (gold) as the drug delivery interface; ii) a near-field communication (NFC) module for the wireless control of triggering signals in controlling location and schedule of drug release. A flexible printed circuit board (PCB) provides interconnected traces that integrate the two parts to form a fully wireless, wearable drug delivery system. The main body of the drug delivery interface (MN arrays) uses poly(D, L-lactide-co-glycolide) (PLGA, Mw = 50 - 75 kDa, ester terminated, Sigma Aldrich) as the matrix materials that can undergo bulk erosion upon contact with biofluids to generate biological benign byproducts (lactic acid and glycolic acid)[53]. As shown in Fig. 1b, the fabrication relies on sputter deposition to coat a thin layer of gold (thickness 150 nm) onto the drug-loaded MN arrays supported by a PLGA base. The gold encapsulation layer is supported on the surface of solid MNs, which enables sufficient stability with a thinner thickness (150 nm) compared with those used in the reservoir designs (thickness of gold layer more than 300 nm) as reported previously (as illustrated in Supplementary Fig. 1). Thus, the thickness to realize effective drug encapsulation can be smaller. Ablation using a laser beam defines the gold traces that connect with separated MN domains to realize spatial control of drug release. A thin polydimethylsiloxane (PDMS) (thickness 10 µm) covers the gold layer at the circuity base regions. It exposes only the MN regions to allow physical contact of the gold layer on the MN with biofluids.

Fabrication of the SOP relies on a low-cost solution-molding procedure (Supplementary Fig. 2). The process starts with UV laser ablation to define an MN mold from a PDMS pad. Drop casting drug-loaded PLGA solution into the PDMS mold and setting under vacuum allow the entrance of PLGA into the negative MN molds. Sufficient solidification over 8 h allows easy extraction of the PLGA MNs. Then, the PLGA MNs undergo sputtering deposition to coat a 150 nm-thick gold layer that conformally covers the top surface, followed by patterning with a laser ablation system to define control circuits. Drop casting a thin PDMS layer (thickness 10 µm) onto gold layers protects the control circuits and exposes the MNs for drug release. The MN patch is then attached to a flexible PCB and connected with a current regulator, a wireless energy harvester, and a microcontroller to complete the fabrication process. The solution-molding method can produce SOPs with: (1) tunable MN lengths from 600 µm to 3 mm and aspect ratio from 3 to 8; (2) high dimension uniformity in both length and base diameter (Supplementary Fig. 3); (3) arbitrary MN array configurations (square, hexagonal, single-needle, etc.). Figure 1e–h

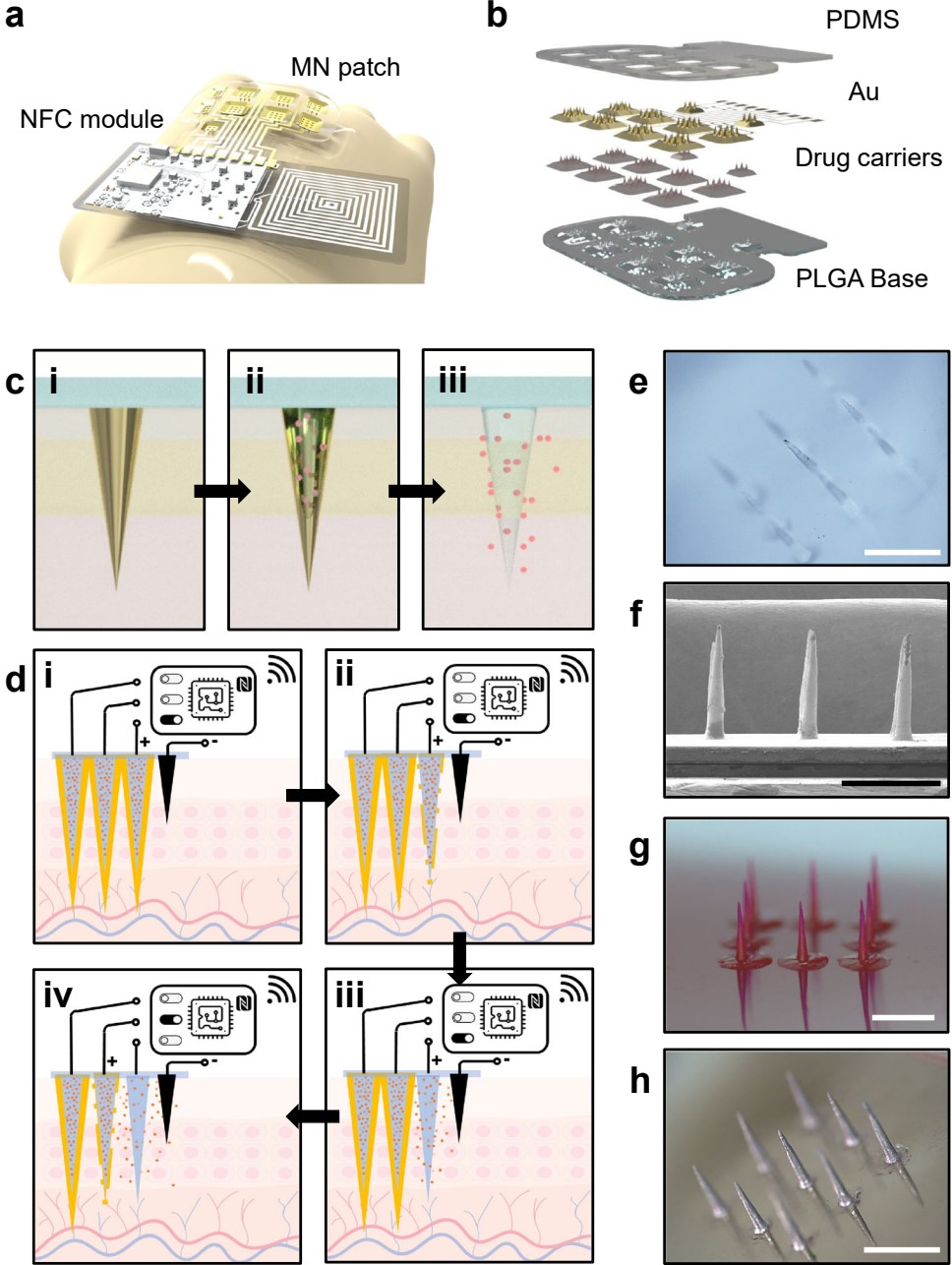

**Fig. 1 | Spatiotemporal on-demand patch for wireless, active control of drug delivery. a** Schematic illustration highlighting the construction of a wirelessly controlled spatiotemporal on-demand patch (SOP) for high-precision drug delivery. The SOP features two main components: (i) an array of drug-loaded microneedles protected by active encapsulation that exploits electrochemically triggered crevice corrosion, for on-demand drug delivery; (ii) a near-field communication (NFC) module assembled on a soft printed-circuit board, for wireless control. **b** Exploded view of the drug-delivery interface of the SOP, including a PDMS encapsulation, an electrically triggerable gold (Au) coating, drug-loaded microneedles based on poly(D, L-lactide-co-glycolide) (PLGA), and a PLGA substrate. **c** Schematic illustration showing process of electrically controlled on-demand drug delivery from an individual microneedle. (i) Standby stage where an encapsulation layer protects the microneedle from releasing drug. (ii)

Transitioning stage where an electrical trigger initiates crevice corrosion of the encapsulation layer to expose drug-loaded base. (iii) Releasing stage, where the exposed base starts to release drugs. **d** Schematic illustration demonstrating the capability of spatiotemporal control of releasing profile from the SOP. (i) Deploying an SOP at the skin interface. (ii–iv) Communicating with the NFC module of the SOP enables active control of drug release for each individual microneedle. **e** Optical image of a PLGA microneedle array. **f** Corresponding SEM image with a tilted view on the PLGA microneedle array. **g** Optical image of a PLGA microneedle array loaded with Rhodamine B. **h** Optical image of a PLGA microneedle array protected with an electrically triggerable encapsulation (Au, thickness 150 nm). The length and base diameter of the microneedles in **e**–**h** is around 1.2 mm and 270 μm, respectively; 3 experiments are repeated with similar results for each in **e**–**h**; the scale bar in **e**–**h** is 1 mm.

shows that the morphology and shape of the PLGA MNs remain stable during both gold deposition and drug loading procedures.

Figure 1c, d illustrates the overall working mechanism to realize high-precision drug delivery by our SOP. The electrically triggered crevice corrosion of the gold protective layer is the switch for initiating

drug release of specific collections of SOP MNs. Upon SOP deployment on the skin, the MNs stay at a standby stage with the drug fully protected by the gold layer. Once a direct current (DC) electrical trigger (2.2–3 V) is applied, the electrochemical crevice corrosion starts to occur on triggered MNs, transitioning them from standby mode to

releasing mode. After a short period (<30 s when 2.5 V is applied), the gold protective layer on the MN is fully dissolved, and MNs are exposed to the bio-environment to enable drug release. The compatible integration of microcontrollers allows the electrical triggers to be manipulated at a precise time point for precision delivery. By patterning gold layers via microfabrication, the SOP can realize the spatial profile of drug release at a high spatial resolution (-1 mm²). A built-in anode is integrated into the SOP to complete the circuitry for the in vivo electrochemical crevice corrosion.

## Characterization of electrically triggered crevice corrosion

Figure 2 demonstrates the characterization of SOP on its active control of drug release (based on MNs, 1.2 mm in height, 150-nm gold coated). The operation of drug-release control includes three stages: (1) standby stage when the MNs are fully coated with gold (labeled as 0 min); (2) transitioning stage when the electrical trigger is activated, and the gold layer is partially dissolved (labeled as 0.5 min); (3) releasing stage when the gold is fully dissolved, and MNs are exposed to the biofluids (labeled as 2 min). 1X Dulbecco's phosphate-buffered saline (DPBS, Corning) is used here to simulate the body biofluid. The experiment is conducted at room temperature and triggered by a 2.5-V DC. Figure 2a shows optical and SEM images of the MN arrays at different stages, which indicates a noticeable change in surface color and roughness associated with the electrically triggered crevice corrosion. The results demonstrate that the main structure of the MN stays stable during the transitioning stage, and the electrical triggers effectively dissolve gold into biofluids and sufficiently expose the core MNs. More characterizations from different perspectives appear in Supplementary Fig. 4.

The electrochemical crevice corrosion of the gold layer can be triggered by a direct current potential within 2 min, which is of clinical relevance for a timely response in drug administrations. A quantitative amperometry study of current density under different potentials is conducted to understand the electrical triggering behavior. Figure 2e shows I-V measurements of SOP triggering with potentials bias ranging from 2.0 V to 2.8 V applied to the MN arrays (1.2 mm in length, 150-nm gold coated). A steep drop in current density appears at 15 s of application for 2.8 V bias, indicating the endpoint of the electrochemical crevice corrosion. The time of the current-drop appearance increases as the potential bias decreases. The time consumption from the beginning to the end of the current is defined as the effective corrosion time plotted against the potential in Fig. 2f. The fitting is based on the Butler-Volmer equation

$$I = i_{corr}[\exp(-\sigma nF\eta/RT) - \exp((1-\sigma)nF\eta/RT)] \qquad (1)$$

where $i_{corr}$ is the exchange current density, $F$ is the Faraday constant, $\eta$ is the overpotential, $R$ is the ideal gas constant, $T$ is the thermodynamic temperature, and $\sigma$ is a coefficient with values ranging from 0 to 1, $n$ is the number of electrons in the anodic half reaction.

This equation describes the relationship between potential difference and reaction rate[54]. At an ambient potential of 2.5 V, the crevice corrosion on MN is triggered within 30 s. This parameter is applied in the following experiments. We hypothesize that the crevice corrosion driven by constant voltage includes two parts: anodic oxidation and mechanical crevices. The 2.5-V potential between the anode and cathode is sufficient for triggering gold oxidation coupled with hydrogen emission reaction (HER) at ambient conditions (1X DPBS, 25 °C). It is reported that an anodic potential larger than 1.95 V (vs standard hydrogen electrode, SHE) is high enough to drive the below-surface oxidation of gold film[55]. This corresponds with our observations, as the crevice corrosion is triggered at a potential above 2.0 V, even though the HER on the counter electrode is not under the standard condition. The oxidated gold species in neutral and alkaline environments may include oxides, hydroxides, and free elements,

mostly in the form of nanoparticles (NPs)[56,57], that collectively in the amount used here are benign to the human body[58]. Supplementary Fig. 5 provides a schematic illustration of the two-electrode system of SOP (Supplementary Fig. 5a) and the atomic scale illustration of the anodic oxidation (Supplementary Fig. 5b, c) on Au (111) facet in a neutral environment, with reaction products, Au₂O₃ in the form of nanoparticles. The anodic behavior is also studied by finite element analysis (FEA), including reaction potential, reaction rate, current, and potential distribution (Supplementary Fig. 6a–d). The simulation results show that the reaction rate at 2.5 V is 281 nm/min (Fig. 2h, i), which is consistent with the experimental observations (150 nm in ~ 30 s). The anodic polarization curve under the condition of standard potentials and primary current distribution also shows a minimal triggering potential at around 2.0 V (Supplementary Fig. 6e). However, the anodic oxidation may not account for the entire loss of the gold coating. Based on the 2.6-V amperometry experiment (Fig. 2e), the theoretical weight of the oxidated gold is calculated as 122 μg. The actual value should be lower because water splitting (oxygen generation), as the side reaction, may also contribute to the current density. This amount of gold NPs does not constitute a health hazard since it is much lower than the safe exposure threshold, which is reported as 5 mg/ml[58]. Meanwhile, the total weight of the diminished gold film on the MN array is calculated as 290 μg, which is significantly higher than the oxidated amount. This is because the major part of the gold layer is not oxidated but exfoliated from the surface due to crevices and cracks. These defects come from the weakening effect on gold membranes as they become thinner during corrosion. This kind of crevice on metal film is generated by an electrical current, which is reported in some previous studies[43,45].

A gold-on-wafer experiment validates the presence of mechanical crevices and exfoliations. The surface roughness of the crevice corrosion of the gold layer is also studied with the gold-on-wafer ({100} facet, 150-nm, 1 cm² square) model (Supplementary Fig. 7b, c). The gold layer is connected to a power source, and crevice corrosion is triggered at 2.5 V in 1X DPBS. The surface roughness of gold is calculated with FIJI ImageJ on optical images collected at various stages of corrosion. As shown in Supplementary Fig. 7d, the surface roughness is monitored every 0.5 min for 5 min from the initial stage, then every 1 min from the 5-min stage to the 9-min stage. A sharp increase in surface roughness is observed in the first 1 min right after the crevice corrosion is initiated (Supplementary Fig. 7a). This is consistent with the amperometry study that most electrochemical corrosion happens within 1 min under this potential. The roughness increase indicates the gold layer's exfoliation from the wafer, which is also observed in the MN corrosion from an SOP. It can be concluded that a significant part of the gold layer is not directly oxidated but exfoliated during the electrical triggers. This corresponds with the previous study using gold film as the control gate for implanted drug reservoirs[44].

To characterize the electrical trigger process, we select two different areas on an SOP MN: the tip area and the waist area, as shown in Fig. 2d. The structural difference between the two areas is the surface curvature, with the former being 1.37 mm⁻² and the latter close to 0. The waist area stands for the MN's major surface, where most drugs are released. The energy-dispersive X-ray spectroscopy (EDXS) mapping is carried out on three stages of MNs from both areas as a semi-quantified surface analysis (SI Figs. 8 and 9). Oxygen, carbon, and gold are selected as elements of interest, where oxygen and carbon show the exposed polymer body while gold shows the encapsulated surface of the MN. The ratio quantification is based on the relative weight percentage of only these three elements despite the existence of other elements. As shown in Fig. 2b, c, both the tip and the waist areas show similar trends in element-relative ratios. An increase in oxygen, carbon, and a decrease in gold (all by weight%) are observed. The waist area, which stands for the main body of an MN, shows a more remarkable change in element constituents, indicating a complete

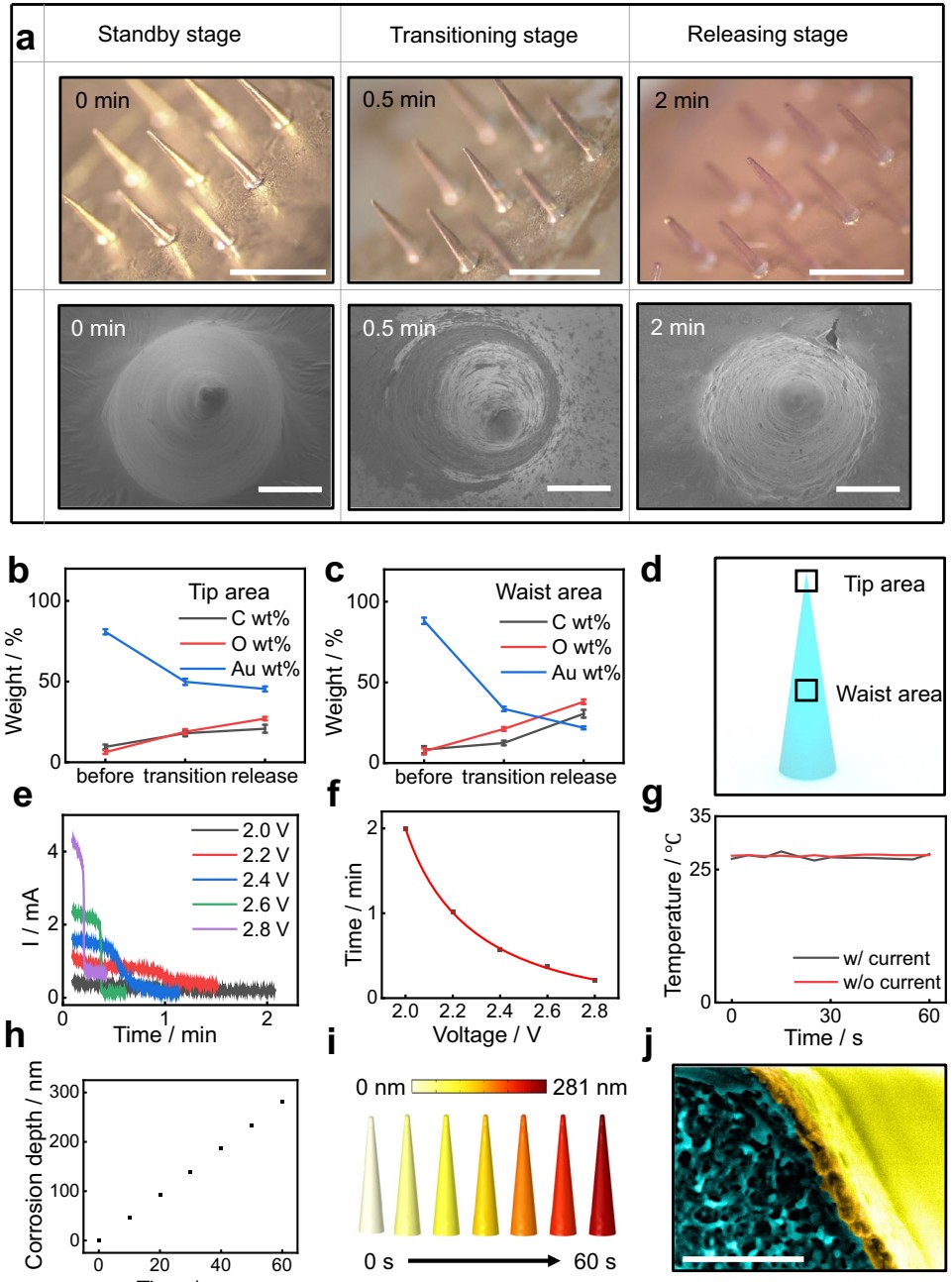

**Fig. 2 | Electrically triggerable encapsulation for active control of drug release.** **a** Optical images and the corresponding SEM to demonstrate an SOP undergoing the process of electrically controlled crevice corrosion. Initially (0 min), the drug-loaded microneedles are fully protected by a layer of gold (150 nm). At 0.5 min, an electrical trigger (2.5 V, in 1X Dulbecco's phosphate-buffered saline) activates the crevice corrosion of the gold encapsulation layer. Then (2 min), the exposed drug embedded inside the microneedle core diffuses to the fluidic environment. Scale bar: 1 mm in optical images, 75 μm in SEM images. **b**, **c** Element analyses by energy-dispersive X-ray spectroscopy (EDXS) of the three stages (as shown in **a**) of the on-demand releasing process of the drug: **b** tip areas of the microneedle; **c** waist areas of the microneedle. Data in **b** and **c** are presented as mean values +/− standard deviation of element weight% from iterations of analysis (*n* = 2). **d** Schematic illustration indicating the corresponding areas of a microneedle analyzed by EDXS

element mapping. **e** Amperometry characterization of the crevice corrosion process of the gold layer (150 nm) coated on microneedles with 1.2 mm height. **f** Measured relationship between corrosion time and trigger potential applied on the gold layer (150 nm), based on the amperometry curve in **e**. **g** Analysis of thermal effect during crevice corrosion process. The comparison between two conditions (with and without current) indicates negligible amount of heat generated in the system. **h** The simulation of crevice corrosion depth on microneedle under 2.5-V in 1X Dulbecco's phosphate-buffered saline. **i** The corresponding corrosion profile on a microneedle in 1 min (2.5-V, 1X Dulbecco's phosphate-buffered saline) related to **h**. **j** The SEM image of the cross-section of an MN (150 nm gold coated, 1.5 mm). Gold surface, gold cross-section, and PLGA cross-section are colored in yellow, orange, and light blue, respectively. 3 measurements are repeated with similar results to **j**. Scale bar: 1 μm. Source data are provided as a Source Data file.

corrosion of triggered crevice for the gold layer. The tip part of MN is still partially capped by a small amount of gold at the releasing stage. This part of the gold layer is isolated during the crevice corrosion process and accounts for the rather high remaining gold shown in

Fig. 2b. The left gold covering the tip area is too small to hinder the overall drug release from the entire MN. Silicon and copper are also observed in EDXS mapping, which comes from PDMS residuals and conductive wires used during the experiment. More detailed

information appears in the Supplementary Information on the EDXS element mapping.

Figure 2g shows the thermal characterization of SOP during the electrochemical corrosion to validate the thermal safety. The experiment uses an MN array (1.2 mm in length, 150-nm gold coated) connected to a 2.5-V DC power to undergo crevice corrosion in 1X DPBS. A FLIR thermometer records the temperature of the MN array with and without current in a 25 °C environment. The results show no noticeable change in temperature during the electrochemical corrosion, which indicates a low possibility of tissue damage from heat effects.

Figure 2j and Supplementary Fig. 10 show the uniformity of the gold layer on MN prior to crevice corrosion. We also use atomic force microscopy (AFM) to characterize the surface roughness of gold layer deposited on polymer to illustrate the uniformity of deposition thickness by sputter coating, as shown in Supplementary Fig. 11.

## Release control and electrical triggering of microneedle patch

Figure 3a demonstrates the encapsulation performance of gold coating on MNs. Here, Rhodamine B (Thermo scientific), a fluorescent dye, is loaded into the microneedle patch (0.3% in weight, ~90 ng per MN) to simulate small-molecule drugs, which can be subsequently quantified by UV-Vis spectroscopy. Figure 3a, b show the release profile of Rhodamine B from a bare MN patch (1.2 mm in length) and an MN patch (1.2 mm in length) with a 100-nm gold coating. The release study is carried out in 45 °C 1X DPBS. Significant color fading on the MN patch without gold coating, as shown in Fig. 3a, and the apparent increase in absorbance (blue, without gold coating, Fig. 3b) indicate a successful release of dye into the biofluids. The average release rate of Rhodamine B in an hour is calculated as ~ 415 ng/min. In contrast, the controlled experiment (black, with gold coating) shows no significant change in absorbance (Fig. 3b), suggesting excellent protection of encapsulated drugs from release. The calculation of the accumulative release amount is based on a calibration curve of Rhodamine B solutions and Beer-Lambert Law. Detailed explanation is provided in Supplementary Fig. 13 and SI Table 1. We further analyze the stability of gold coating in a 0.5% agar model to better simulate the mechanical properties of animal tissue. The MN array (1.2 mm in length, 150-nm gold coated) remains stable during a 2-week soaking test in the agar model without significant changes in shape or surface morphology (Fig. 3a). SI Figs. 14 and 15e, together with Fig. 3a, show the stability of gold film on MNs against soaking in biofluids and mechanical friction in tissues. This indicates that the bonding between gold and PLGA is strong enough to meet the need for implantation, which corresponds with previous research on Au-polymer adhesion[59-61].

Figure 3c demonstrates the wireless design of a SOP. An external power source—in this case, a signal generator—is connected to an inductive coil to provide a high-frequency (~MHz level) alternative current (AC). The inductive coil is paired with the receiving coil on the device to achieve wireless power transfer via magnetic resonance coupling. The AC is then converted to direct current using a full-bridge rectifier. A 2.5-V regulator then regulates the DC to provide a stable potential that facilitates the electrochemical crevice corrosion of the gold layer on MNs. The configuration of the wireless SOP is illustrated in Fig. 3d. The wireless SOP consists of an energy harvesting module for wireless power, a power amplification module, a System-on-Chip (SoC) module for remote control, and an MN array coupled with counter electrodes (CE). Within the energy harvesting module, a receiver coil, a full-bridge rectifier, and a regulator are used to provide stable direct current (DC) output. The output is coupled with a 0.1-μF capacitor to constitute a low-pass filter, improving output stability. An equivalent circuit diagram is provided in Fig. 3g, which emphasizes the energy harvesting module while simplifying the power amplification and SoC. A complete circuit diagram can be found in Supplementary Fig. 16. Supplementary Fig. 17 shows the performance and impedance analyses of the energy harvesting module. The optimal signal input is

determined as around 15 V peak-to-peak at 15 MHz. The measurements shown in Fig. 3e validate the output stability of wireless SOP with the regulator. The final output power signal is stabilized at around 2.5 V with a standard deviation of ~0.1 V, ensuring precise crevice corrosion control for initiating drug release. The input signal's optimal frequency is around 36 MHz (10 V peak-to-peak signal), as shown in Fig. 3f.

## Temporal and spatial control of SOP triggering

To further investigate the temporal and spatial controllability of electrical triggering on the MN patch (Fig. 4a), we design a multi-domain SOP to realize the stepwise, on-demand release. The multi-domain SOP consists of 7 domains of MN arrays (1.2-mm in length, 150-nm gold coated). The gold layer on the PLGA patch is patterned by laser ablation to enable separate triggering of individual MN domains. A layer of PDMS (~10 μm) is then applied onto the patch except for the hexagonal MN regions to protect gold interconnects from dissolving during electrical triggering. Figure 4c shows the electrical triggering schedule for four of the seven MN domains for the SOP loaded with Rhodamine B (0.3% by weight) during its immersion in 65 °C 1X DPBS as an accelerated study. The electrical triggers use 2.5-V DC bias for 30 s at every 30-min interval of immersion. Following each interval, immediate sampling of the environmental fluids allows quantitative estimation of drug-release dosage using UV-Vis spectroscopy, as shown in Fig. 4b. The measurements show a multi-step increase in spectral absorbance, indicating the stepwise increase of drug dosage and confirming the on-demand drug release at the desired time (0, 30, 60, 90 min, Fig. 4b). Figure 4d demonstrates the staged release of MN domains by selectively dissolving the gold encapsulation layer with an electrical trigger (Supplementary Movie 1). The red dash line circles the specific MN array triggered from each stage. The results confirm that our SOP realizes both temporal and spatial control of drug release using digital electrical triggers.

Supplementary Fig. 18 demonstrates the capability of ultrafine spatial control of drug release. Here, we design a miniaturized SOP with a single domain consisting of 8 MNs (1.1 mm in length, coated with 150 nm-thick gold) with 1−3 mm in spatial separation. With a specific design of gold circuits, each MN in the domain can be triggered individually (as illustrated in Supplementary Fig. 18a). Supplementary Fig. 18b shows that each MN follows sequential release via electrical triggering (2.2-V DC) within 15 s without interfering with adjacent MNs. Patterning techniques on the gold layer primarily dictate the spatial resolution of release control for SOP.

## Demonstration of SOP in intracranial drug delivery

Beyond the clinical applicability for transdermal drug delivery, the SOP could show impactful utility in facilitating animal behavior studies. Here, we demonstrate intracranial delivery of melatonin using SOP for an animal sleep study. Melatonin, a hormone naturally produced in the brain by the pineal gland, plays a crucial role in regulating the sleep-wake cycle while also participating in other regulations like blood pressure and body temperature[62-64]. However, people found many strains of nocturnal laboratory mice are deficient in the release of melatonin, including C57/B6 mice[65,66], and the effects of melatonin are still controversial[67-69]. Here, we use C57/B6 mice and implant the SOP to test how exogenous spatiotemporally controlled release of melatonin in the deep locations of the cortex, which is mainly distributed by melatonin receptor 1[70], would affect the sleep-wake cycle. The high spatiotemporal controllability of melatonin release offered by the SOP may open up new opportunities to understand regional brain responses to melatonin and study the pathology of sleep-wake disorders. Loading melatonin into MNs of the SOP follows the solution fabrication method described in Supplementary Fig. 2. Melatonin that dissolves in acetone can be mixed with the precursor PLGA solution, which ensures the loading dosage (10−35 wt%) for each MN. The 3-mm MNs are fabricated with the PLGA-melatonin ratio from 10:1 to 2:1

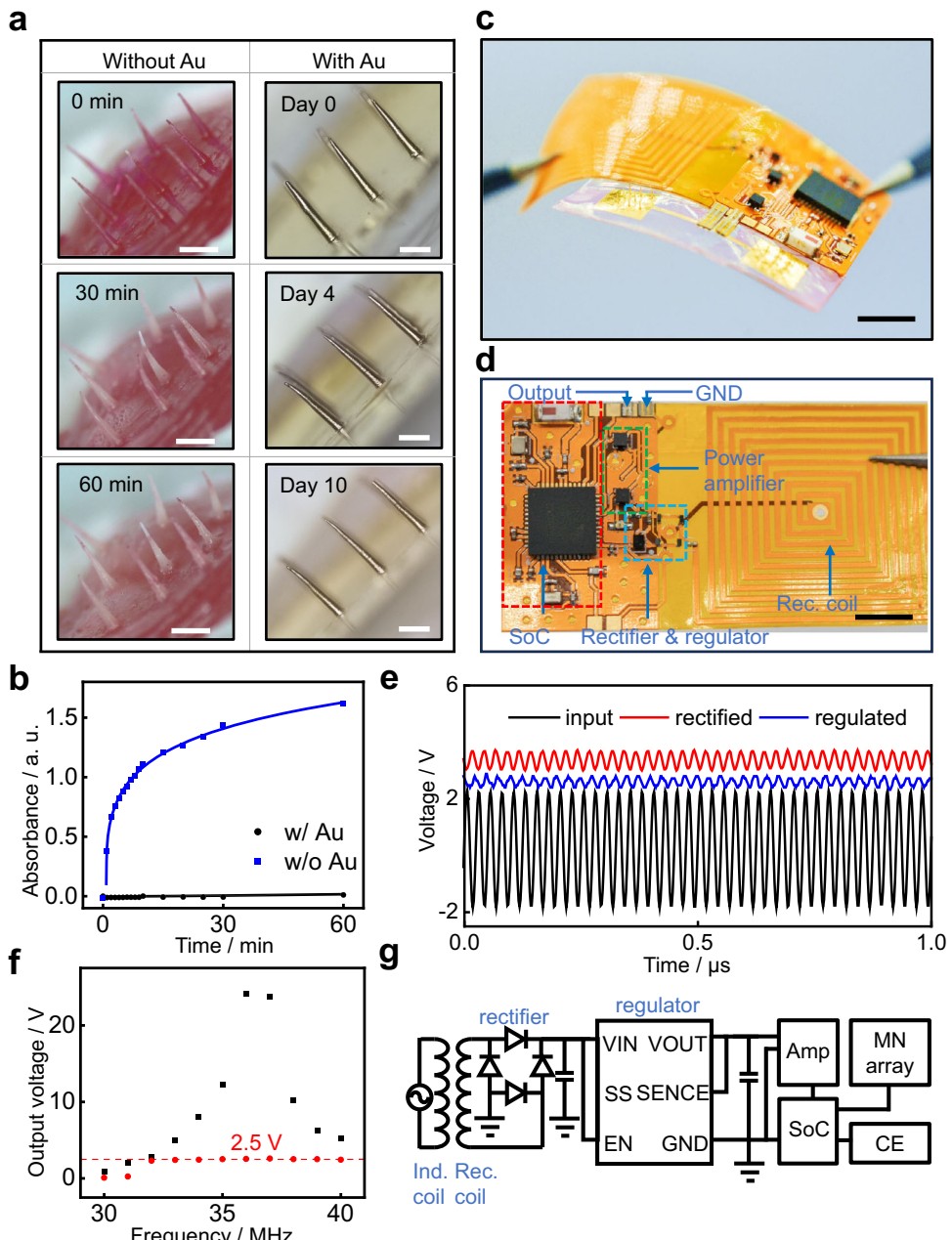

**Fig. 3 | Wireless drug release control of SOP via near-field communication.**
**a** Optical images of microneedles with and without gold coating, respectively. The gold layer remains stable in an artificial tissue (0.5% agar in 1X Dulbecco's phosphate-buffered saline, PBS) for more than 10 days with no observable degradation. Scale bar: 500 µm. **b** The release profile of Rhodamine B loaded in microneedles with and without gold coating. Height of the microneedles, 1.2-mm; Thickness of the gold coating, 100 nm. All samples were immersed in 1X Dulbecco's PBS at 45 °C. **c** Optical image of a wireless SOP. Scale bar: 5 mm. **d** Top-down view of the wireless module used in the SOP, featuring the receiving coil (labeled as Rec. coil), the full-bridge rectifier and the regulator (circled in blue), the power amplifier (circled in green), the System-on-Chip (SoC) module (circled in red), and the ground (GND) & output channels. Both GND and output channels are connected to gold-coated MN arrays via conductive traces, which serve as a counter electrode

and a working electrode, respectively. Scale bar: 5 mm. **e** Measurement on power harvesting of the wireless SOP (without SoC module). Black, input AC signal (peak-to-peak 4 V, 39 MHz); Red, rectified received signal (3.4 V, stand deviation (s.d.) - 0.2 V); Blue, regulated output signal (2.6 V, s.d. - 0.1 V). **f** The frequency matching characterization of the wireless power transfer. Black dots: the output voltage after rectification without regulation; Red dots: the output voltage after rectification and 2.5-V regulation.10-V peak-to-peak input signal is used, and the working range from 32–40 MHz is determined at 36 MHz. The potential rectification achieves stable bias around 2.5 V as the electrical trigger. **g** The circuit diagram of the energy harvesting module of SOP with correspondence to **e** and **d**. Inductive coil (Ind. coil) is not present in **c** and **d**. The power amplification and SoC module are abbreviated as Amp and SoC. CE stands for counter electrodes. Source data are provided as a Source Data file.

(Fig. 5a). In addition, the drug can be concentrated around the tip of the MN, which ensures deep-brain delivery for mice, as shown by the dash-red line in Fig. 5a. Based on the mixing ratio, the payload of melatonin per MN is estimated to be 22.2 to 81.7 µg, comparable to recommended dosages for mice (4–20 mg/kg)[71,72]. The drug payload is

modulated by loading different PLGA solutions during the mold casting procedure. Furthermore, the mechanical property of the 3 mm-tall MNs used for intracranial delivery is characterized by a fracture test (Supplementary Fig. 15). The ultimate strength of PLGA MN (1.2 mm in length) is determined by the first fracture point in the force-

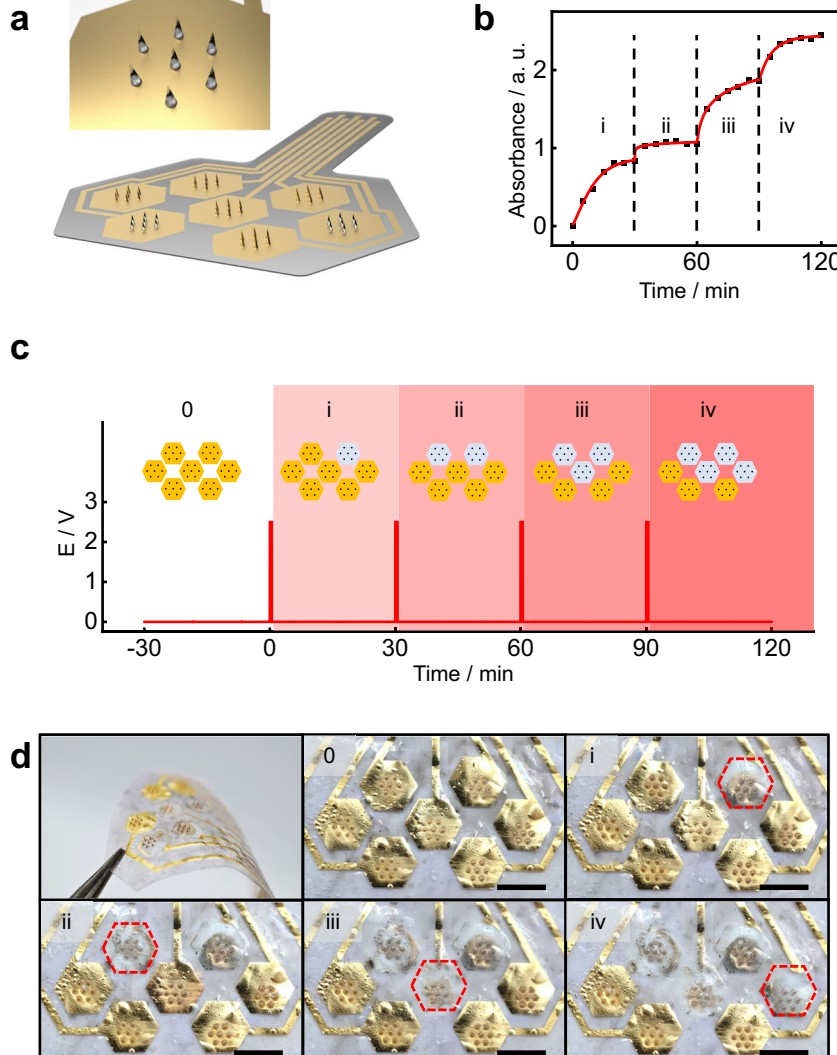

**Fig. 4 | Characterization of the spatiotemporal control of drug release.**
**a** Schematic illustration of a multi-domain SOP, with a zoom-in view of a micro-needle domain at the releasing stage. **b** The stepwise release profile of a multistage drug release simulated by Rhodamine B: the four-step release of Rhodamine B is electrically triggered at 0, 30, 60, and 90 min, corresponding to labels i, ii, iii, and iv, respectively. The SOP demonstrated here has microneedles with a height of 1.2 mm, coated with a 150-nm gold layer, and protected by a 10-μm PDMS layer. **c** Schematic illustration showing the sequential electrical-triggering schedule on the multi-array SOP. The electrical triggering uses a DC voltage of 2.5 V for ~ 30 s. **d** Optical images of the multi-domain SOP undergoing a sequential electrical trigger. The domain triggered at each stage is labeled by red hexagonal dashed frames. The images from Stage 0 to Stage iv correspond to **b** and **c**. 2 experiments are repeated with similar results to **d**. Scale bar: 5 mm. Source data are provided as a Source Data file.

displacement graph, as labeled by the red frame (Fig. 5b). The fracture point, recognized by a sudden drop in measured force, corresponds to the initial fracture of the MNs, followed by multiple subsequent fractures appearing in different locations of the MNs (Supplementary Fig. 15b, d). The maximum mechanical strength is derived to be 118 MPa, which is sufficiently rigid for human skin penetration[73]. The calculation considers the pressure measured at the first fracture point, while the contact area of the MN is approximated based on the tip diameter (around 30 μm, Supplementary Fig. 3f). Another concern is the stability of the gold layer during implantation. Supplementary Fig. 15e shows the PLGA MN (150 nm gold coated, 1.2 mm) before and after the penetration test. No significant changes can be observed based on the comparison of optical images. Figure 3a also proves the stability of the gold layer during the soaking test. In addition, we use the MN array (150 nm gold coated, 1.2 mm) to penetrate the tissue of the chicken thigh for multiple cycles, as presented in Supplementary Fig. 14. The gold layer remains stable after 20 cycles of penetration.

Deployment of the melatonin-loaded SOP in live animal models as they move in a caged environment shows possibilities for actively controlling melatonin release to the deep-brain regions of the parietal lobe (Fig. 5g). To ensure device stability, the SOP is coupled with a custom headstage that can be firmly mounted onto mice heads (Supplementary Fig. 19). Figure 5h shows an immunohistochemistry analysis of brain tissues from the mice at various recovery stages following SOP implantation. On day 1 of post-implantation, the brain tissues in close proximity to the SOP MN show structural damage resulting from mechanical forces during intracranial surgery, which is typical for general brain implantation[74]. As the mice recover from the implantation, the levels of GFAP (red) and IBA (white) show a significant decrease in concentration and staining range, indicating excellent biocompatibility of SOP. The tissue in contact with the MN becomes smoother, with fewer rough edges. An obvious increase in neuron regeneration (neurotrace, green) can be observed, indicating good recovery from implantation surgery.

Supplementary Fig. 20 shows an in vivo study to validate the safety and biocompatibility of SOP in intracranial deployment. Here, the study compares four groups of MNs (bare MN (MN), gold-coated MN (Au-MN), melatonin-loaded MN (Mel-MN), and melatonin-loaded

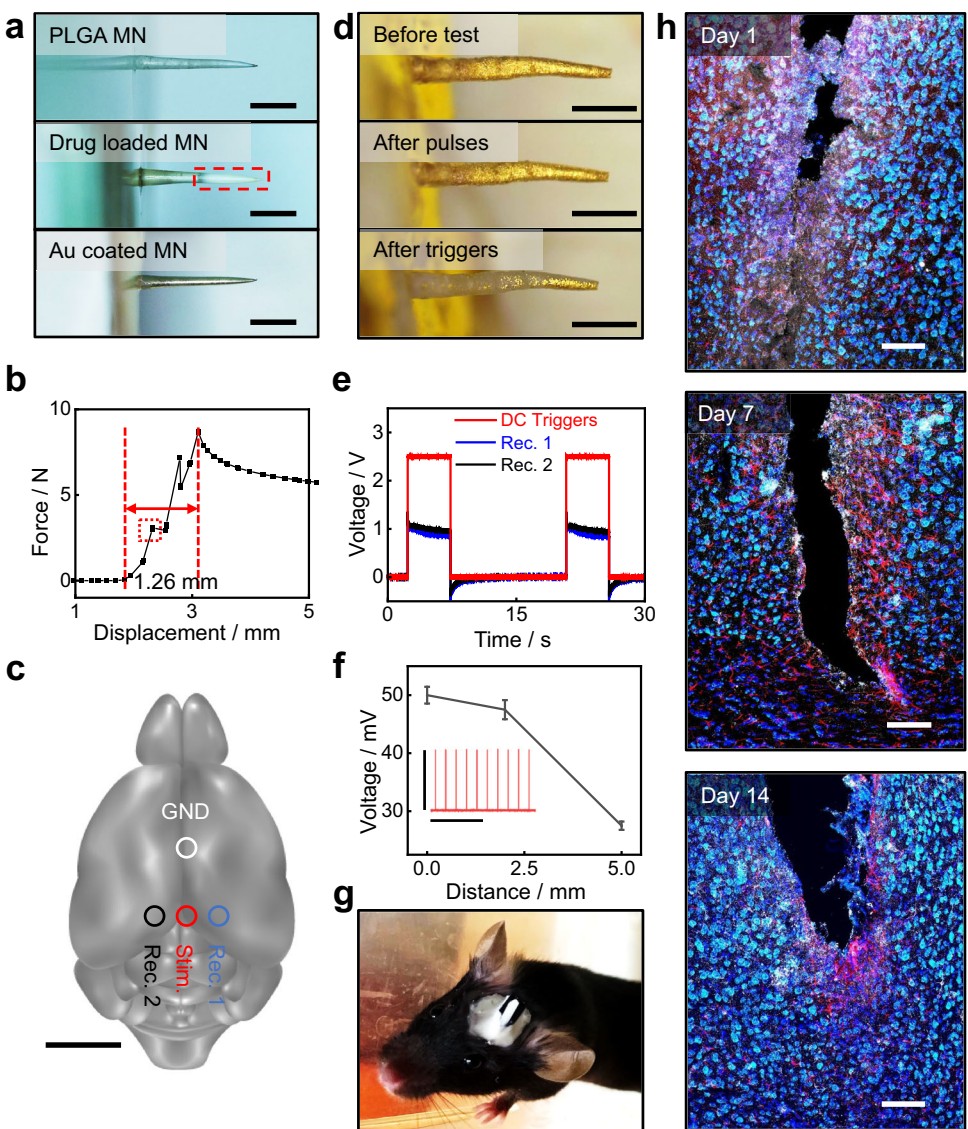

**Fig. 5 | In vivo demonstration of SOP. a** Optical images of an intracranial micro-needle (height, 3 mm) at various fabrication stages: (1) PLGA needle base; (2) loading melatonin (20%); (3) coating with 150-nm gold. The drug-concentrated region is circled with a red frame. (3) measurements are repeated with similar results to **a** Scale bar: 1 mm. **b** During a fracture test, the measured force-displacement curve of a microneedle array (9 needles, 1.2 mm in length). A red frame circles the first fracture point, and the displacement in contact is labeled and measured as 1.26 mm. **c** The schematic illustration of a mouse brain model indicating the deployment location of SOP in the in vivo study. The red and white circles label the positions of the SOP microneedle and counter electrode (Pt wire); the blue and black circles label the positions of two separate recording electrodes (Pt wire). Scale bar: 5 mm. **d** Optical images of the microneedle (3 mm in length) before the test, after the pulse trigger, and after the square wave trigger, respectively. 3 measurements are repeated with similar results to **d**. Scale bar: 1 mm. **e** Measured square-wave triggers generated from the SOP microneedle at various distances to the microneedle. The gold-coated microneedle delivers the trigger (3-mm, 150-nm) as a 2.5-V 5-s square wave periodically. The recording electrodes (Rec. 1 and Rec. 2) are around 5 mm from the microneedle. **f** Measured pulsatile triggers generated from the SOP microneedle at various distances to the micro-needle. A pulse signal (50 mV, 1 Hz, 10 ms in width) is applied by a microneedle electrode and recorded by a Pt wire at ~ 0.1, 2.0, and 5.0 mm, respectively. The signal for stimulation is attached in red. Scale bar: 5 s, 50 mV, for the $x$ and $y$ axis, respectively. Data are presented as mean values +/− standard deviation of peak amplitude ($n = 10$) **g**. Image of the freely moving mouse after SOP deployment. **h** Immunohistochemical staining images of the recovery process after microneedle implantation. The combined images include Nissl bodies (neurotrace, green), astrocytes (glial fibrillary acidic protein (GFAP), red), activated microglia (lba1, white) and nuclear DNA (4',6-diamidino-2-phenylindole (DAPI), blue). Scale bar: 100 μm. ($n = 3$ independent experiments) Source data are provided as a Source Data file.

MN with gold coating (Au-Mel-MN)) in their effects on the inflammation and degree of neural damage after one-month implantation. Supplementary Fig. 21a shows typical images of MN implanting sites with GFAP/IBA/NSE/DAPI multiple staining. As a result, the number of astrocytes and microglia in the region of interest (ROI) is significantly lower in the Au-MN or Au-Mel-MN group compared to the bare MN group, which indicates that gold-coated MNs have less damage to neural tissues and cause less inflammation compared to bare PLGA MNs. In addition, the fluorescence density of the markers has no difference between bare and gold-coated MNs, which indicates that the Au encapsulation on MNs does not affect the expressing strength of the GFAP or IBA signals (Supplementary Fig. 21c).

Furthermore, we used a fiber photometry system to investigate the neural activity in the medial prefrontal cortex (mPFC) after the implantation of MNs (bare MN, Au-MN, control (no MN), respectively, Supplementary Fig. 22). As shown in Supplementary Fig. 22c, there is no observable difference in $Ca^{2+}$ dynamics of mPFC excitatory neurons between the control group, bare MN, and Au-MN groups, which

indicates the MNs have no significant damage to the brain tissue and introduce little influence on the neuronal activity.

Besides, we study the bioresorbability of the MN. Though the degradation of PLGA is much slower than gold[75], we show this process of PLGA in an accelerated degradation experiment in 65 °C phosphate-buffered saline (PBS), as illustrated in Supplementary Fig. 23.

We further use the in vivo animal models to characterize the SOP functional performance. Here, an additional two recording electrodes are inserted adjacent to the location of SOP implantation, as shown in Fig. 5c. First, a set of DC electrical triggers (5 s duration, 2.5 V) is delivered and recorded (5 mm from the stimulation electrode), as shown in Fig. 5e. The crevice corrosion of the gold layer (thickness 150 nm) on the MN can be completed by applying 5–7 times the 5-s triggers to allow melatonin release. Then, a series of short pulse stimulation experiments are examined (Supplementary Fig. 24). The pulse signals vary from 10 mV to 50 mV in amplitude and 1 Hz (duration 10 ms) or 10 Hz (duration 1 ms) in frequency. The signal amplitude-recording distance relationship is also studied based on 10-ms pulses of 50 mV (Fig. 5f) and proved to affect a 5-mm area mainly. No significant changes are observed on the gold layer of MN after short pulses, as demonstrated in Fig. 5d, indicating electrical signals generated from neuron cells induce negligible damage to the gold protection layer of the SOP. Furthermore, the results demonstrate the gold-coated MNs can also be used as stimulation electrodes for the delivery of low-amplitude (10–50 mV), pulsatile signals (Fig. 5f), which may serve as a strategy for neuronal regeneration[76,77].

## Assessment of in-brain drug delivery by microneedles

To investigate the gold encapsulation's performance in controlled drug release from the MN, we fabricated two groups of melatonin-loaded (25 wt%) MN with the exact specification shown in Fig. 5a. For effective comparison, the first batch of MNs (Mel-MNs) has no gold encapsulation layer, while the second batch of MNs (Au-Mel-MNs) is coated with 200 nm of gold on the MN surface.

First, we conduct the in vivo drug release experiment with the Mel-MNs and the blank control (MNs without loading melatonin) by stereotaxically implanting the MN into the medial prefrontal cortex (mPFC) of mice (one MN in each, Fig. 6a). After one week of recovery, we use the Pinnacle sleep system (Sirenia Acquisition) to record EEG and EMG analyses. From 19:00 (active period) to 22:00, the power density of the Mel-MN group is compared with those of the control group (Fig. 6b). The total amount of NREM sleep increased by 42.0%, and the amount of wakefulness decreased by 27.9% over the 3-h period compared to these parameters in the control group (Fig. 6c, d). The sleep status is calculated from the EEG/EMG traces and the corresponding hypnograms (Supplementary Figs. 25–29). The experiment confirms the effective release of Melatonin from Mel-MNs, which can successfully modulate the sleep behavior of mice over a week.

Second, we conduct similar in vivo experiments to investigate the feasibility of release control with gold encapsulation from the Au-Mel-MN. Here, we apply the Au-Mel-MNs by stereotaxically implanting them into the medial prefrontal cortex (mPFC) of mice (one MN in each, Fig. 6e). For effective comparison, the EEG and EMG recording starts one week after implantation, following the same way as described above. Figure 6f shows the power density of the Au-Mel-MN group and control group recorded from 19:00 to 01:00 am (active period), indicating that the delta power density of the Au-Mel-MN group was significantly higher ($P < 0.01$) than that of the control group. The significantly enhanced delta band EEG power indicates a deeper sleeping degree induced by melatonin, compared with the Mel-MNs experiment (Fig. 6c, d). The total amount of NREM sleep increased significantly by 36.7%, REM sleep increased by 57.8%, and that of wakefulness significantly decreased by 26.0% over the 6-h period, compared to these parameters in the control group (Fig. 6g, h). Essentially, melatonin release in the mPFC induces NREM sleep

amount increases in active periods and enhances delta power density during NREM sleep even more significantly than that shown in the Mel-MN group. The delayed release effect mainly explains this due to the existence of gold encapsulation. As a result, the releasing peak of melatonin appeared after the controlled degradation of the gold protection layer. In contrast, the Mel-MN group releases melatonin right after implantation, leading to a weakened effect at the time point of recording (one week post to recovery). Collectively, these in vivo experiments further demonstrate the programmed drug release performance of SOPs in freely moving animals and suggest its potential utility in animal behavior studies. We revealed that programmed drug release of exogenous melatonin in the mPFC would improve NREM sleep, REM sleep, and delta power density, which may represent a novel target for treating sleep disorders.

In this study, we present an on-demand drug-delivery patch that can be digitally controlled to enable delivery precision in both space and time. The spatiotemporal on-demand patch (SOP) uses bioresorbable microneedles with high-aspect ratios (3–8) as the drug-loading vehicle and a thin layer of gold (thickness 150 nm) as a release gate that can be digitally controlled with a small electrical trigger (2.5 V for 30 s). This design allows fully active control of drug release at a single-microneedle level with spatial resolution, demonstrated here, less than 1 mm², highlighting the enabling capabilities over most existing drug-delivery devices. This spatial resolution allows more than 20 doses of the drug to be housed within a thin, wearable patch of 1 cm², ensuring comfortable and convenient user adherence for repetitive pharmaceutical treatment over extended periods. The fabrication is compatible with the microfabrication process, which could further decrease the spatial resolution to micrometer scales. The on-demand, rapid-response drug release can be enabled within 30 s following an active electrical trigger.

The fabrication procedure of our SOP uses a simple solution-molding method, offering a low-cost approach with high dimensional customizability. Using laser ablation, microneedles, ranging from 0.6–3 mm, can be manufactured efficiently and in high quality to fit a wide range of drugs (melatonin with controllable payloads[78]. Moreover, the solution-based mold process allows drug-load PLGA MNs to be produced at scale with similarly high quality. The process allows convenient compatibility for integrating electronic modules to enable digital automation in drug delivery.

The multifunctionality of drug delivery and stimulation therapy could synergistically create advanced therapy as potential future avenues of neuroscience research. The SOP presented in this work exhibits continuous sustained drug release feature that meets the need to treat various chronic neural diseases apart from sleep disorder, especially Alzheimer's disease (AD)[50]. As a multiphase neural disease that affects most areas of the brain[51,52], the treatment of AD requires different drugs at different stages. For example, acetylcholinesterase inhibitors (AChEIs) are currently the mainstay of treatment for mild-to-moderate AD. In contrast, memantine is approved for mild-to-moderate dementia as a non-competitive N-methyl-D-aspartate (NMDA) receptor antagonist that prevents the overactivation of neuronal NMDA receptors. At the same time, the combination of these two drugs also significantly improves cognitive function in moderate-to-severe AD[79]. The SOP can enable precise joint delivery of multiple drugs in desired brain regions to achieve enhanced treatment. Combined with a burst release design (like hollow microneedles), the SOP also fits for rapid drug delivery as a timely response towards acute neural diseases, such as cataplexy and epilepsy[80–82]. The remotely powered SOP allows for active drug delivery at a particular time and space coupled with real-time behavioral study, such as sleep-quality characterization, which provides a convenient method for drug-performance analysis. The high-resolution and controllability features of SOP make it suitable for such kind of regional and dosage-sensitive treatment. In addition, many drugs for brain disorders may

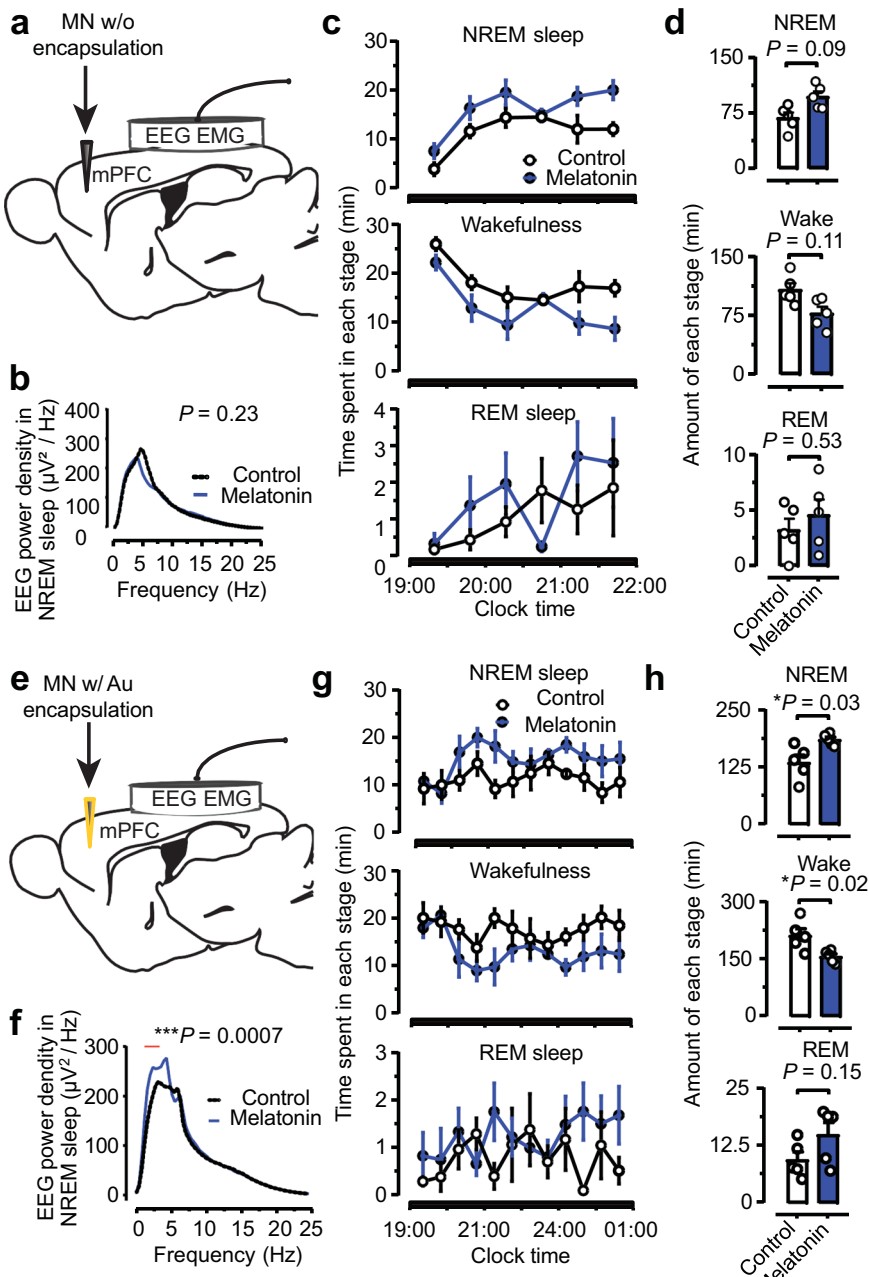

**Fig. 6 | Assessment of in-brain drug delivery by microneedles.** The drug delivery analysis of melatonin-loaded microneedle (Mel-MN) is illustrated from **a**–**d**, and melatonin-loaded microneedle with 200-nm gold encapsulation (Au-Mel-MN) from **e** to **h**. **a**, **e** Schematic illustration of recording brain states by implanting the microneedle and electrodes after 1 week in mice. The microneedle is implanted into the medial prefrontal cortex (mPFC) of mice brain, while electrodes for signal recording are chronically implanted through the skull (for electroencephalogram, EEG) or into trapezius muscles (for electromyography, EMG). The microneedle in **a** and **e** are both loaded with 25 wt% melatonin (Mel-MN), while the microneedle in **e** is further coated with a 200-nm gold layer (Au-Mel-MN). **b**, **f** EEG power density analysis. The recording is a 3-h period from 19:00 to 22:00 in **b**–**d** and a 6-h period from 19:00-1:00 in **f**–**h**. **c**, **g** Time course of different brain states (active period) immediately after the light turns off. The lengths of non-rapid eye movement

(NREM) sleep, wakefulness, and rapid eye movement (REM) sleep are calculated every 30 min. Two groups of 5 mice (melatonin-delivered mice and the control, respectively) are compared, where melatonin-delivered mice showed longer NREM sleep and shorter wakefulness in both **c** and **g**. The implanted Au-Mel-MN showed a more substantial effect of melatonin, suggesting the delayed release of melatonin as a result of gold protection. **d**, **h** Duration of different brain states (NREM, wake, and REM) during the recording time, corresponding to **c** and **g**, respectively. Data in **c**, **d**, **g**, **h** are presented as mean values +/− SEM of peak amplitude ($n = 5$ in all melatonin loaded and control groups) Paired two-sided $t$-tests are used for comparisons between the two groups in **d**, **h**, **b**, and **f**. Two-way ANOVAs are used to perform group comparisons with Sidak's multiple comparisons test in **c** and **g**. All statistical tests were two-tailed. Source data are provided as a Source Data file, where time duration in **c**, **d** and **g**, **h** is counted in half-minute units.

have detrimental effects on other organs if administered systemically[83]. The regional drug release feature of SOP enables targeted drug delivery that minimizes systemic exposure, reducing the risk of unwanted side effects in non-brain tissues.

Moreover, our SOP is capable of wireless operation via near-field communication or, potentially, Bluetooth Low Energy. The agar-soaking test, fracture test, and in vivo intracranial delivery experiments validate the SOP's practical functionality and biocompatibility.

Furthermore, gold-coated MNs can extend beyond drug-releasing control, as the SOP can offer in vivo electrical stimulation. These concepts establish unique approaches in high-precision drug-delivery technologies with additional utilities in advancing fundamental studies of disease pathology (such as cancer metastasis) and neuroscience research, as demonstrated by both benchtop experiments and in vivo studies. Future efforts on fully digital automation of drug delivery will pave the way for the next generation of precision medicine.

## Methods

The sample size used in this study is based on the expected variations between animals and is comparable to many previous reports using similar techniques (cited in the corresponding sections). Sample size of each experiments can be found in the Figure legends. Data from animals were excluded based on histological criteria that included injection sites, virus expression and optical fiber placement. Only animals with injection sites/virus expression/optical fiber placement in the region of interest were included, based on our previous reports. The experiments were not randomized. Animal were allocated into experimental groups by matched gender, age, weight, etc. Investigators were blinded to the experimental groups until all data had been collected and analyzed.

C57BL/6 mice (8–10 weeks, females) were used for all animal experiments. Animals were group-housed at constant temperature (22–24 °C) and humidity (40–60%), and bred in a dedicated husbandry facility with 12/12-h light-dark cycles with food and water ad libitum and under veterinary supervision. Animals subjected to surgical procedures were moved to a satellite housing facility for recovery with the same light-dark cycle. All procedures were conducted in accordance with the NIH Guide for the Care and Use of Laboratory Animals and with the approval of the Institutional Animal Care and Use Committee at the University of North Carolina at Chapel Hill (UNC), under the protocol # 22–146[84].

### Fabrication of the SOP

**Normal MN patch.** The general procedure of MN fabrication was demonstrated in Supplementary Fig. 2. First, 50 g of PDMS (Sylgard 184, Dow Corning) was fully cured in a glass petri dish at 60 °C for 2 h with 5 g of its corresponding curing agent. Then, a negative MN mold was patterned on the cured PDMS by a UV laser ablation system (SFX-5UV, Luoyang Xincheng Precision Machinery). The MN molds were fabricated with different depths from 0.5 to 3.5 mm, a base diameter of around 0.25 mm, and an inter-needle spacing of 1 mm. The depth of the MN mold can be controlled by tuning the loops and power of UV laser ablation. The UV ablation was followed by acetone sonication for at least 5 min to clean up the surface of the PDMS negative mold. Then, a PLGA (PLGA, Mw = 50 - 75 kDa, ester terminated, Sigma Aldrich) solution (10 wt% in acetone, VWR) was drop cast on the PDMS negative mold in the petri dish. The PLGA-covered mold was heated at 45 °C and 60–160 mmHg for around 2 min to let the PLGA solution fill in the mold and evaporate. The entire PDMS mold was capped by another petri dish to slow down the evaporation of acetone. The evaporation process was followed by a refill of PLGA solution. The evaporation-refilling cycle was conducted 10–20 times to provide enough PLGA for the MN patch, with a thickness from 0.6 to 1.2 mm. After that, the PLGA-covered mold was kept in the oven at 45 °C and 1 atm for at least 8 h to dry the surface. The mold was then frozen at −20 °C for at least 30 min to harden the PLGA patch, which was subsequently extracted from the mold. The free-standing PLGA patch was allowed to further dry up on both sides at 45 °C and 1 atm for another 24 h, then trimmed by UV laser ablation. The hardened and dry PLGA patch was eventually deposited with a layer of gold (usually 150 nm in thickness) by sputter coating (PVD 75 sputterer, Kurt J. Lesker). The Gold traces were patterned by an IR laser ablation system (SFX-50GS, Luoyang Xincheng Precision Machinery).

**Melatonin-loaded MN.** A melatonin loaded PLGA solution was prepared in advance. The melatonin (Sigma Aldrich) was mixed with PLGA (Sigma Aldrich) at a ratio by weight from 1:10 to 1:2. Then, the mixture was dissolved in acetone (VWR) at a 1:10 ratio by weight. The solution is stored in refrigerator at around 4 °C no longer than 12 h, and wrapped with aluminum foil to avoid light.

First, 50 g of PDMS (Sylgard 184, Dow Corning) was fully cured in a glass petri dish at 60 °C for 2 h with 5 g of its corresponding curing agent. Then, a negative MN mold was patterned on the cured PDMS by a UV laser ablation system (SFX-5UV, Luoyang Xincheng Precision Machinery). The MN molds were fabricated with a depth around 3 mm, a base diameter of around 0.35–0.5 mm, and an inter-needle spacing of at least 5 mm. The depth of the MN mold can be controlled by tuning the loops and power of UV laser ablation. The UV ablation was followed by acetone sonication for at least 5 min to clean up the surface of the PDMS negative mold. Then, the melatonin-PLGA solution (10 wt% in acetone) was drop cast on the PDMS negative mold in the petri dish. The PLGA-covered mold was heated at 30 °C and 60–160 mmHg for around 2 min to let the PLGA solution fill in the mold and evaporate. The entire PDMS mold was capped by another petri dish to slow down the evaporation of acetone. The evaporation process was followed by a refill of PLGA solution. The evaporation-refilling cycle was conducted 10–20 times to provide enough PLGA for the MN patch, with a thickness from 0.6 to 1.2 mm. After that, the PLGA-covered mold was kept in the oven at 30 °C and 1 atm for at least 8 h to dry the surface. The mold was then frozen at −20 °C for at least 30 min to harden the PLGA patch, which was subsequently extracted from the mold. The free-standing PLGA patch was allowed to further dry up on both sides at 30 °C and 1 atm for another 24–48 h, then trimmed by UV laser ablation. The hardened and dry PLGA patch was eventually deposited with a layer of gold (usually 150–200 nm in thickness) by sputter coating (PVD 75 sputterer, Kurt J. Lesker). The Gold traces were patterned by an IR laser ablation system (SFX-50GS, Luoyang Xincheng Precision Machinery).

**Rhodamine B-loaded MN.** A Rhodamine B (Thermo scientific) loaded PLGA solution was prepared in advance. Rhodamine B was dissolved in the acetone (VWR) solution of PLGA (usually at 1/300 ratio by weight, Sigma Aldrich).

First, 50 g of PDMS (Sylgard 184, Dow Corning) was fully cured in a glass petri dish at 60 °C for 2 h with 5 g of its corresponding curing agent. Then, a negative MN mold was patterned on the cured PDMS by a UV laser ablation system (SFX-5UV, Luoyang Xincheng Precision Machinery). The MN molds were fabricated with different depths from 0.5 to 3.5 mm, a base diameter of around 0.25 mm, and an inter-needle spacing of 1 mm. The depth of the MN mold can be controlled by tuning the loops and power of UV laser ablation. The UV ablation was followed by acetone sonication for at least 5 min to clean up the surface of the PDMS negative mold. Then, the Rhodamine B-PLGA solution (10 wt% in acetone) was drop cast on the PDMS negative mold in the petri dish. The PLGA-covered mold was heated at 45 °C and 60–160 mmHg for around 2 min to let the PLGA solution fill in the mold and evaporate. The entire PDMS mold was capped by another petri dish to slow down the evaporation of acetone. The evaporation process was followed by a refill of PLGA solution. The evaporation-refilling cycle was conducted 10–20 times to provide enough PLGA for the MN patch, with a thickness from 0.6 to 1.2 mm. After that, the PLGA-covered mold was kept in the oven at 45 °C and 1 atm for at least 8 h to dry the surface. The mold was then frozen at −20 °C for at least 30 min to harden the PLGA patch, which was subsequently extracted from the mold. The free-standing PLGA patch was allowed to further dry up on both sides at 45 °C and 1 atm for another 24 h, then trimmed by UV laser ablation. The hardened and dry PLGA patch was eventually deposited with a layer of gold (usually 150 nm in thickness) by sputter coating (PVD 75 sputterer, Kurt J. Lesker). The Gold traces were

patterned by an IR laser ablation system (SFX-50GS, Luoyang Xincheng Precision Machinery). For controlled dye release model, a layer of PDMS (1:10, ~ 15 μm) was carefully, manually coated and cured on specific positions that should be covered.

**Wireless on-demand patch.** A commercially available Pyralux Kapton soft PCB material made of polyimide sandwiched between copper was sprayed on and coated with masking paint (Krylon). The paint mask was then partially removed using an infrared laser cutter to expose unwanted copper, which was removed by etching in ferric chloride solution (MG Chemicals 415) for 15 min. The soft PCB was then rinsed with water and acetone to remove the remaining etchant and paint mask. Surface mount electrical components, including diodes and power regulators, were soldered using solder paste and hot air guns. An MN array (1.2-mm, 150 nm gold coated) was integrated into the wireless patch using PI-based adhesives. A small jumper wire was attached to the MN array using silver-based conductive adhesives to ensure conductivity. Another MN array was integrated into the wireless patch in the same way to serve as the counter electrode of crevice corrosion.

### Kinetic characterization of electrochemical corrosion
An MN patch coated with gold was connected to a piece of graphene tape. The peripheral area of the MN patch (except for needles) and the graphene tape (5113SFT, 3 M) were protected by PDMS from corrosion, with an opening area of 0.5 * 0.5 cm$^2$. An amperemeter (NI-USB 4065, National Instruments) was applied to monitor the current density versus time. The experiment was carried out in a two-electrode system, where the counter electrode was graphene tape. A power source (SPD3303X-E, Siglent) was used to provide a constant voltage between the cathode and anode, while the anode was connected to the gold-coated MN patch to provide oxidative potential. The MN patch with graphene tape was coated with PDMS (~10 μm in thickness) to protect the exposed surface except for the MN regions. The electrochemical corrosion was carried out in a standard environment (1X DPBS, Cornings) to mimic the body fluid. I-t curves were obtained at 2.2, 2.4, 2.6, 2.8, 3.0 V.

Kinetic characterizations were also carried out for Mo-coated MN arrays in the same way (Supplementary Fig. 12).

### Study of dye release
**Free release without encapsulation.** Rhodamine B (Thermo scientific, mass ratio versus PLGA 1:300) was mixed with PLGA (Sigma Aldrich) and dissolved in an acetone (VWR) solution. The fabricated MN patch was immersed in a petri dish containing 20 mL of deionized water (HAVENLAB) at 45 °C constant temperature. Samples were taken for UV-Vis spectrometry since the MN patch is immersed every 1 min from 0 to 10 min, every 5 min from 10 to 30 min, and at 60 min. The UV-Vis absorbance was characterized by a UV-Vis spectrophotometer (VWR-10037, VWR) from 800 to 300 nm, with an interval of 1 nm. The samples were returned to the petri dish immediately after characterization to maintain a constant volume. The absorption data was obtained by VWR UV software and visualized by Origin Pro 2022.

**Encapsulated release with encapsulation.** A Rhodamine B-loaded MN patch was deposited with a 100-nm gold layer on the side with needles to encapsulate the PLGA and dye. The back side of the MN patch was fixed and encapsulated into PDMS to prevent exposure to water. The patch was immersed in a petri dish with 20 mL of 1X DPBS (Cornings) at 45 °C. The UV-Vis absorbance was characterized by a UV-Vis spectrophotometer (VWR-10037, VWR) from 800 to 300 nm, with an interval of 1 nm. The samples were returned to the petri dish immediately after characterization to maintain a constant volume. The absorption data was obtained by VWR UV software and visualized by Origin Pro 2022.

### On-demand stepwise release.
A 150-nm gold layer was deposited to a Rhodamine B-loaded MN patch by sputter coating on the side with needles, then patterned by IR laser ablation to generate gold traces. The gold electrodes were connected by silver paste (8331D, MG Chemicals) to the constant voltage power source. Then, the entire PLGA patch was encapsulated with PDMS except for the needle region. The patch was immersed in a petri dish with 20 mL of 1X DPBS at 65 °C. A 2.5 V constant voltage was applied to trigger the electrochemical corrosion of gold within 20 s. The dye release of the four MN arrays was subsequently triggered every 30 min. The UV-Vis absorbance was characterized by a UV-Vis spectrophotometer (VWR-10037, VWR) from 800 to 300 nm, with an interval of 1 nm. Samples were taken at certain intervals for UV-Vis absorbance characterization and returned to keep the volume constant. The absorption data was obtained by VWR UV software and visualized by Origin Pro 2022.

### Surface profilometry of corrosion
A silicon wafer ({100} facet, UniversityWafer) was deposited with a 150-nm gold layer by sputtering and divided into 4 cm$^2$ squares. The gold-coated side of the wafer was connected to the DC power source by graphene tape. The peripheral area of the wafer square was encapsulated by PDMS with a 1 cm$^2$ window exposed in the center, as illustrated in Supplementary Fig. 7. The wafer square was immersed in 10 mL 1X DPBS (Cornings), and the electrochemical corrosion was triggered by a 2.5 V constant voltage. Optical images of the wafer square were captured by a microscope (S9i, Leica) starting from the beginning. The time interval of images was 0.5 min from 0 to 5 min and 1 min from 5 to 9 min. The obtained images were cropped to leave only the exposed window in the center and further analyzed by FIJI ImageJ (Java 1.8.0_172). Arithmetic mean roughness (Ra) and root mean square roughness (Rq) were calculated for each image by the roughness analysis module.

### Study of thermal effect
A 150-nm gold-coated MN array was placed in a Petri dish and immersed in 10 mL 1X DPBS (Cornings) at room temperature (25 °C). The MN array was soldered with a copper wire and connected to the DC power source. The infrared radiation image was recorded by a thermal camera (ETS320, FLIR). For the first 1 min, the temperature of the MN array was recorded without any voltages applied. For the following 1 min, the MN array was applied with a 2.5 V constant voltage. Temperature data points were taken every 5 s from the videos.

### Elemental analysis of corrosion
The electrochemical corrosion of MN arrays (1.2 mm in length, 150-nm gold coated) was conducted at 2.4 V in 1X DPBS (Cornings). The triggering time was 0.5 min and 2 min for two MN arrays, representing the "transitioning" and "release" stages of the corrosion procedure. Optical and SEM images were captured, as shown in Fig. 2a.

The EDXS element mapping was conducted by the Scanning Electron Microscope (SEM, Hitachi S-4700). 20 kV was applied under analysis mode at a selected 100 * 120 μm$^2$ area, and the signal was collected for 400 s. For MN samples at the releasing stage, which have little gold coating left, a 5-nm Pd layer was sputter coated before characterization to increase the surface conductivity. The data analysis was automatically done by the INCA software (Oxford Instruments).

### Mechanical strength measurement
A force measurement system (Mark 10, ESM 303) was used to study the mechanical strength of MNs. An MN array (9 or 25-needle, 1.2-mm, w/ or w/o 150-nm gold coated) was attached to the top sample holder by glue and double-sided tapes (Supplementary Fig. 15g). The bottom sample holder was placed with a piece of glass to serve as a hard object. Both the glass and the MN array were placed as horizontally as

possible. The sampling rate of the force gauge was set as "as high as possible", and the moving speed of the sample holder was set as 13 mm/min, which was the lowest value. The sample holder will gradually descend to a point where the MNs were in contact with the glass and stopped manually when the MN array is fully crushed (Supplementary Fig. 15a, c). Another soft contact experiment was also conducted under the same parameters except for the glass, which was replaced by 0.5% agar to mimic the brain (Supplementary Fig. 15e–f).

## Wireless power transfer analysis

A signal generator (SDG2042, Siglent) was used to provide a radio frequency (RF) signal to the inductive coil. Performance at several different frequencies (30–40 MHz) is examined to find the optimal transmission efficiency. The high-frequency alternating current is then fed into the inductive coil closely coupled with the receiving coil, which is connected to a rectifier and a power regulator to provide a DC voltage for MNs (1.2 mm, 150 nm gold coated) immersed in saline solution. The voltage is measured using a benchtop oscilloscope (SDS1204, Siglent). The impedance of the MNs is measured using a portable network analyzer (NanoVNA). The power dissipated by the MNs during the electrochemical process, ($P_{needle}$), can be calculated using both the voltage measurement (peak-to-peak amplitude, $V_{pp}$) and the real part of the impedance under the frequency at which measurements (Re($Z_{needle,frequency}$)) are performed using the following equation:

$$P_{needle} = \frac{\left(\frac{V_{pp}}{2} \cdot \frac{\sqrt{2}}{2}\right)^2}{\text{Re}\left(Z_{needle,frequency}\right)} \qquad (2)$$

## Finite element analysis of crevice corrosion

The finite element analysis (FEA) of gold layer crevice corrosion on MNs was simulated by COMSOL 6.1. A model MN pair (1.2 mm, 3.5 mm in the distance) was set up to simulate the exposed surface of cathode and anode in the fluid. The tip of the MN was treated as a hemisphere (50 μm in diameter) to facilitate convergence, while the bottom face was set as a circle (270 μm in diameter). The surface of one MN was set as the cathode boundary, and the other was set as the anode boundary, as shown in Supplementary Fig. 6b–d. The entire simulation domain was set as a cubic (5 mm in length) space, including the two MNs. To simulate the physical and chemical properties of 1X DPBS, water (Water, liquid (mat1)) from the built-in database was chosen as the material of the cubic domain (not including MNs). The electroconductivity was set to be 1.6 S/m, which was a typical value provided by the manufacturer of PBS. Other surfaces of the model (except for the MN surfaces) were set as insulation in the boundary conditions. The stationary and transient simulations were performed based on secondary current distribution, which considered overpotentials while assuming homogeneous electrolytes. The anode equilibrium potential, according to literatures[85], was set as 1.83 V (Au$^+$|Au) to describe the electrochemical force needed to trigger gold oxidation. The cathode equilibrium potential was set as 0 V, and the electromotive force, which was the voltage of the external power source, was set as 2.5 V unless specified. The model was automatically meshed at the ultrafine level under physics-controlled mode. The current density distribution (Supplementary Fig. 6a) and iso-potential surface (Supplementary Fig. 6b–d) were simulated at the starting point of the crevice corrosion, which was stationary. The anodic polarization curve applying different electromotive forces (2.0–3.2 V) is simulated, as shown in Supplementary Fig. 6e. Transient simulation is conducted for the corrosion depth versus time, assuming the gold layer on the MN is thick enough. The corrosion depths from the starting time (0 s) to 60 s are monitored every 10 s (Supplementary Fig. 6f) and visualized (Supplementary Fig. 6g).

## Immunohistochemical analysis

Mice were given a lethal dose of pentobarbital sodium (Sigma Aldrich), followed by intracardial perfusion with 4% paraformaldehyde (Sigma Aldrich) in PBS, as reported previously[86–88]. Then, the brains were dissected, post-fixed for 24 h at 4 °C, and cryoprotected with a solution of 30% sucrose (Fisher Scientific) in 0.1 M phosphate buffer (pH 7.4) (Sigma Aldrich) at 4 °C for at least 24 h, fully submerged. This was followed by cutting into 40-μm sections, washing three times in PBS, three 5-min incubations in 1 mg/ml$^{-1}$ sodium borohydride (Honeywell Fluka) in PBS, then 1-h incubations in 1% Triton-X-100 (Sigma Aldrich) in PBS. A blocking step was then performed using 5% donkey serum (Sigma Aldrich) in 0.3% PBST (phosphate-buffered saline with Triton X-100) for 1 h. Brain sections were then incubated for ~16 h at 4 °C in a blocking buffer containing goat anti-GFAP of Santa Cruz Biotechnology (1:1000, Cat# sc-6170) and rabbit anti-Iba1 of Fujifilm Wako (1:500, Cat# 019-19741). Sections were then transferred to a secondary antibody solution containing Fisher Scientific Alexa Fluor 647 donkey anti-rabbit IgG (1:1000, Cat# A32795), Alexa Fluor 568 donkey anti-goat IgG (1:1000, Cat# A11057) and Neurotrace 435/455 Blue Fluorescent Nissl stain (1:100, Cat# N21479) in 0.1% PBST for 1 h at 24 °C, with intermittent brief periods of shaking. Sections were washed three times for 30 min each in 0.1% PBT, with 1 μM DAPI (Invitrogen) solution included on the third wash step. After rinsing, slices were dried on a piece of slide glass and coverslipped. All brain slices were imaged with an Olympus FV3000 microscope and obtained by FV31s-VW software. All images were processed with the same settings using the Fiji software by ImageJ (Java 1.8.0_172).

## Stereotaxic surgery

Mice were anesthetized under 1.5–2% isoflurane in oxygen (Baxter Healthcare) at 0.8 LPM flow rate. Probes covering melatonin or vehicle were implanted unilaterally into the medial prefrontal cortex (anteroposterior (AP): −1.5 mm, mediolateral (ML): ±0.3 mm, dorsoventral (DV): −2.1 mm). Then, mice were chronically implanted with EEG/EMG electrodes for polysomnographic recordings. The electrodes consisted of two stainless steel screws connected to EEG Teflon-coated wires, which were inserted through the skull, and two EMG Teflon-coated wires that were bilaterally placed into both trapezius muscles. All of the electrodes were fixed to the skull with dental cement and attached to a microconnector[89]. The scalp wound was closed with surgical sutures, and each mouse was kept in a warm environment until it resumed normal activity as previously described[90,91].

## Polygraphic recordings and vigilance state analysis

All mice were recorded by an EEG/EMG polysomnographic system of Pinnacle. The recordings of EEG and EMG were performed by means of a specially designed slip ring so that the behavioral movement of the mice would not be restricted. First, cortical EEG and EMG signals were amplified and filtered (EEG, 0.5–30 Hz; EMG, 20–200 Hz) and then digitized at a sampling rate of 200 Hz and recorded by using Sirenia Acquisition (Pinnacle). When completed, polygraphic recordings were automatically scored off-line by 4 s epochs as wakefulness, REM, and NREM sleep by SLEEPSIGN (Kissei Comtec, Nagano, Japan) according to standard criteria[92]. As a final step, defined sleep-wake stages were examined visually and corrected, if necessary. The data related to EEG/EMG were analyzed by GraphPad Prism 8 and Matlab R2022b.

## Gold layer thickness analysis

To analyze the uniformity of the gold layer, we prepare two MN samples: (1) a 1.5 mm MN with 150 nm gold layer is broken in half after freezing (Supplementary Fig. 10a–c); (2) a 1.2 mm MN with 100 nm gold layer undergoes partial Au film exfoliation (Supplementary Fig. 10d–f). By SEM, the cross-section images of the MN reveal the detailed structure of the outer gold layer. Supplementary Fig. 10c shows the cross-section of the gold layer (colored orange), which is

approximately estimated at 160 nm in width. Considering the existence of a diffusion layer (where gold atoms mix with polymer), the error in thickness is acceptable. Namely, Supplementary Fig. 10f shows the thickness of the exfoliated gold film, roughly calculated to be ~ 95 nm, after perspective correction. Both samples show good uniformity of the thickness of the gold layer, which provides good encapsulation performance. Furthermore, we utilize AFM to characterize the surface roughness of the coated gold layer on the polymer substrate, as presented in Supplementary Fig. 11. The average and root-mean-square (RMS) roughness is calculated to be less than 3 nm, which meets the requirement of thickness uniformity.

## Immunohistochemical analysis (additional)

Brain sections were then incubated for ~16 h at 4 °C in a blocking buffer containing goat anti-GFAP of Santa Cruz Biotechnology (1:1000, Cat# sc-6170), rabbit anti-Iba1 of Fujifilm Wako (1:500, Cat# 019-19741), and Chichen anti-Neuron Specific Enolase (1:100, NSE Millipore Cat# AB9698).

All data values were presented as the mean ± standard deviation (SD). For analyzing fluorescence density and mean of peak value in the region of interest (ROI), we define the ROI as the areas within 200 μm lateral to the edge of the probes. One-way ANOVA was used to compare the fluorescence density and mean of peak value between different experimental groups.

As is well known, GFAP and IBA-1, representing astrocytes and microglia, respectively, reflect the inflammation in the brain tissues[93,94]. Neuron-specific enolase (NSE), an acidic protease unique to neurons and neuroendocrine cells and a sensitive indicator for assessing the severity of nerve cell damage and prognosis, is widely used to reflect neural injuries and pathological processing[95].

## Stereotaxic surgery (additional)

For in vivo fiber photometry recordings, mice were unilaterally injected with 250 nl of AAV5-CaMKII-GCaMP6f mixed with 5 nl AAV5-CaMKII-tdTomato (from UNC Vector Core) into the mPFC at these coordinates AP: −1.5 mm, ML: +0.3 mm, DV: -2.1 mm. Optical fibers (Newdoon Inc, O.D.: 1.25 mm, core: 200 mm, NA: 0.37) were implanted 0.2 mm above the mPFC (AP: -1.5 mm, ML: +0.3 mm, DV: -1.9 mm). At the same time, MN probes were implanted ipsilaterally into the medial prefrontal cortex at 30 degrees lateral to the optical fiber.

## Fiber photometry system

The multi-fiber photometry system was used as previously described[96,97]. Briefly, the system consisted of a 488 nm excitation laser, a fluorescence cube, and a spectrometer. The 488 nm laser beams first launched into the fluorescence cube and then into the optical fibers. The GCaMP and tdTomato emission fluorescence collected from the fiber probe traveled back to the spectrometer. Only animals with strong GCaMP and tdTomato expression were included in the study (Supplementary Fig. 22a). Spectral data was acquired by OceanView software (Ocean Optics, Inc) at 10 Hz and was synchronized to a 20 Hz video recording system to acquire the animal behavior.

The in vivo recordings were carried out in an open-top home cage (21.6 " 17.8 " 12.7 cm) in the 30 Lux red light environment. Laser power was adjusted to a final optical fiber output of 30 mW. Photometry data were exported to MATLAB R2014b for analysis. Coefficients of GCaMP6f and tdTomato were unmixed by a customized script by fitting spectrum signals to standard emission curves. GCaMP6f signals were normalized by tdTomato signals for motion correction. (Supplementary Fig. 22a). A 0.1 Hz high-pass filter corrected the fluorescence bleaching. Photometry signals (ΔF/F) were derived by calculating (F−F0)/F0, where F0 is the median of the fluorescence signal (Supplementary Fig. 22b). For the home cage analysis, we recorded data for 5 min per mouse and calculated the ΔF/F to further analyze the correlation of SuM and DG signals in raw data (Supplementary Fig. 22c). Cumulative activity = ΔF/F x Time. The average of the peak of ΔF/F above 2 SD during 5 mins in the home cage was calculated.

## PLGA MN accelerated degradation

We conducted an accelerated degradation experiment of PLGA MN (3 mm, without drug payload) by soaking it in 65 °C 1X DPBS (Cornings). We observed changes in the shape and morphology during the immersion (Supplementary Fig. 23). The MN was taken out of the solution for observation after 30 min, 1 h, 6 h, and 24 h. It is clear that, after 24-h degradation, the main body of PLGA MN collapsed. This experiment simulates the chronic degradation of PLGA in the biofluidic environment under ambient conditions, showing significant degradation of PLGA MN.

## Reporting summary

Further information on research design is available in the Nature Portfolio Reporting Summary linked to this article.

## Data availability

The data of this study are available within the article, the Supplementary Information. Source data are provided with this paper and available upon request. The data generated in this study are provided in the Supplementary Information and Source Data files. Source data are provided with this paper.

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

## Acknowledgements

This work was supported by the start-up funds from University of North Carolina at Chapel Hill and the fund from National Science Foundation under award # ECCS-2139659 (received by W.B.). Research reported in this publication was also supported by the National Institute of Biomedical Imaging and Bioengineering at the National Institutes of Health under award # 1R01EB034332-01 (received by W.B.). This work was performed in part at the Chapel Hill Analytical and Nanofabrication Laboratory, CHANL, a member of the North Carolina Research Triangle Nanotechnology Network, RTNN, which is supported by the National Science Foundation, Grant ECCS-2025064, as part of the National Nanotechnology Coordinated Infrastructure, NNCI.

## Author contributions

W.B. conceived and directed the project. Y.W. and Z.C. performed experiments and prepared supplementary information. B.D., W.L., S.X., L.Z., T.W., P.H., Z.Y., and W.X. helped fabricate the device and collect the data. Z.C., Z.H., and J.S. performed the animal experiments. Y.W., Z.C., J.S., Z.H., and W.B. wrote the paper. All authors discussed the results and commented on the manuscript.

## Competing interests

The University of North Carolina at Chapel Hill (no. 63/343,888; filed 18 May 2023) filed a provisional patent application surrounding this work. The method to digitally automate the control of drug delivery is included in the patent. Wubin Bai, Yihang Wang, "Wearable Apparatus for Deep

Tissue Sensing and Digital Automation of Drug Delivery", provisional patent, No. 63/343,888, filed on May 19, 2022. The remaining authors declare no competing interests.
