## [Peer Review File · Nature Communications]

REVIEWER COMMENTS

Reviewer #1 (Remarks to the Author):

We have thoroughly reviewed the manuscript titled "Digital automation of transdermal drug delivery with high spatiotemporal resolution" submitted to Nature Communications. This manuscript mainly focused on investigating a spatiotemporal on-demand electrically-triggered microneedles patch for controllable drug release with high precision in space and time. We regret to think that the current version of the manuscript may not be suitable for publication in Nature Communications. Below are some concerns regarding the study.

1. Figures needs significant reorganizing. Figures were organized in a simple and unreadable way while the figure legends were too tedious, which made this manuscript hard to read. Also, some figure descriptions in the manuscript didn't match the Figures. For example, where is the PBC? I cannot find it in Figure 1a. Besides, most texts in the figures were too small to read out, such as "i, ii, iii" in Figure 1.
2. Authors should explain how the thickness of gold layer was tailored and measured. Also, EDS mapping should be utilized to further ensure the gold distribution.
3. A schematic illustration about how the electrically controlled crevice corrosion of the SOP works should be provided.
4. Safety assessment of SOP should be carried out to obtain a more comprehensive understanding of the potential side effects. This additional research would enhance the credibility and reliability of the findings.
5. The reference part was organized in a poor way. Some lacked the author's full surname and some lacked the specific page numbers or article numbers.
6. The English writing needs improving as there are many grammatical errors over the manuscript.

Reviewer #2 (Remarks to the Author):

This study reports a novel transdermal drug delivery system called the "spatiotemporal on-demand patch (SOP)". The system combines drug-loaded microneedles with biocompatible metal films to achieve digitally controlled drug release with high precision in both space and time. Some concerns should be addressed, as follows:

1. The active release behavior of the SOP system can be effectively initiated by opening the Au film gate by corrosion-induced crevices via electrical triggers. However, in the manuscript, the authors do not give a detailed explanation of the reactions that occur between the anode and cathode. Please provide a more detailed explanation of how the drug delivery system works and how the mechanism of electrically controlled drug release is implemented.
2. Line 75-77, the authors mentioned that a typically used metal gate for forming electronic implants that enable on-demand drug delivery is a gold layer with a thickness of approximately 300 nm. Why does the authors choose a 150 nm thick Au layer in this work?
3. The thickness of the gold layer has a significant effect on drug release. How the authors precisely control the thickness of the Au coating in this work? And what about the uniformity of the Au coating? Could you please provide a cross-section picture of the Au coated microneedles to demonstrate the gold coating thickness on the surface of PLGA microneedles? What about the bonding force between the gold coating and the surface of PLGA microneedles? And is there a risk of the gold layer falling off during the insertion of the microneedles into the skin?
4. Figure 3b suggested that the MN patch with gold coating showed excellent protection of encapsulated drugs from releasing. And the authors mentioned that "The average release rate of Rhodamine B in an hour is calculated as ~ 343 ng/min" (line 314 - 315). How is this result obtained from Figure 3b? In fact, there is a lack of data supporting the drug cumulative release kinetics of both PLGA microneedles and Au-coated microneedles. Please provide the cumulative drug release amount data of both W/ Au and W/O group at different time points based on Figure 3b.
5. Since acetone solvent is used in the preparation of microneedles, it is crucial to take into account the potential safety concerns arising from any residue left by the solvent. Therefore, conducting a cytotoxicity test becomes essential in order to assess the biocompatibility and safety of the microneedles. Please supplement the cytotoxicity test of the MNs system.
6. Is the Au coated PLGA MNs completely degraded and absorbed after administration without removal? How long does it take for PLGA microneedles to degrade in mice?
7. There are some mistakes in figure description. For example, SI Figure 12 shows the representative confocal images of 40- μ m horizontal cortical slices at various stages after implantation of the bioresorbable electrode probes, not "performance and impedance analyses of this wireless SOP". The authors should revise the manuscript in general to correct such errors. In addition, the authors also should check the references for formatting inconsistencies.

Responses to comments of Referee #1

Comments from Referee #1:

Summary Comment: “We have thoroughly reviewed the manuscript titled "Digital automation of transdermal drug delivery with high spatiotemporal resolution" submitted to Nature Communications. This manuscript mainly focused on investigating a spatiotemporal on-demand electrically-triggered microneedles patch for controllable drug release with high precision in space and time. We regret to think that the current version of the manuscript may not be suitable for publication in Nature Communications. Below are some concerns regarding the study.”

Our response: We thank the referee for these insightful comments and for these helpful suggestions for revision. We carefully addressed the issues, as listed below, and revised our manuscript accordingly.

Modification to the manuscript: None.

Comment 1: “Figures needs significant reorganizing. Figures were organized in a simple and unreadable way while the figure legends were too tedious, which made this manuscript hard to read. Also, some figure descriptions in the manuscript didn't match the Figures. For example, where is the PBC? I cannot find it in Figure 1a. Besides, most texts in the figures were too small to read out, such as “i, ii, iii” in Figure 1.”

Our response: We thank the referee for this comment. We adjusted the figure size, format, legends, and descriptions. The order of **Figure 3** and **Figure 5** was significantly changed. In addition, we removed unnecessary titles, legends, and comments or described them in the caption to make the figure more readable. In **Figure 1**, for the term “PBC”, we assume it refers to PCB or printed circuit board. The PCB is indicated as the electronic components in **Figure 1a**. To further illustrate the device, we made a new schematic illustration of **Figure 1a**, as shown below. In this illustration, the SOP device can be divided into two parts: the microneedle (MN) patch and the near-field communication (NFC) module. Regarding the font of the texts in the figures, we made substantial adjustments to enhance the readability. For example, we increased the size of most serial numbers in the figures, including the “i, ii, iii” in **Figure 1**. In **Figure 3**, the images of wireless SOP are updated in **Figure 3c-3d**. We also adjusted **Figure 3g** as a more complete electrical diagram. **SI Figure 16** is also added for a detailed explanation of the electronic design.

Figure 1. Spatiotemporal on-demand patch for wireless, active control of drug delivery. **a.** Schematic illustration highlighting the construction of a wirelessly controlled spatiotemporal on-demand patch (SOP) for high-precision drug delivery. The SOP features two main components: **i)** an array of drug-loaded microneedles protected by active encapsulation that exploits electrochemically triggered crevice corrosion, for on-demand drug delivery; **ii)** a near-field communication (NFC) module assembled on a soft printed-circuit board, for wireless control. **b.** Exploded view of the drug-delivery interface of the SOP, including a PDMS encapsulation, an electrically triggerable gold (Au) coating, drug-loaded microneedles based on poly(D, L-lactide-co-glycolide) (PLGA), and a PLGA substrate. **c.** Schematic illustration showing process of electrically controlled on-demand drug delivery from an individual microneedle. **i)** Standby stage where an encapsulation layer protects the microneedle from releasing drug. **ii)** Transitioning stage where an electrical trigger initiates crevice corrosion of the encapsulation layer to expose drug-

loaded base. **iii)** Releasing stage, where the exposed base starts to release drugs. **d.** Schematic illustration demonstrating the capability of spatiotemporal control of releasing profile from the SOP. **i)** Deploying an SOP at the skin interface. **ii)- iv)** Communicating with the NFC module of the SOP enables active control of drug release for each individual microneedle. **e.** Optical image of a PLGA microneedle array. **f.** Corresponding SEM image with a tilted view on the PLGA microneedle array. **g.** Optical image of a PLGA microneedle array loaded with Rhodamine B. **h.** Optical image of a PLGA microneedle array protected with an electrically triggerable encapsulation (Au, thickness 150 nm). The length and base diameter of the microneedles in **e, f, g, h** is around 1.2 mm and 270 μm , respectively; the scale bar in **e, f, g, h** is 1 mm.

Figure 3. Wireless drug release control of SOP via near-field communication. a. Optical images of microneedles with and without gold coating, respectively. The gold layer remains stable in an artificial tissue (0.5 % agar in 1X Dulbecco's phosphate-buffered

saline, PBS) for more than 10 days with no observable degradation. Scale bar: 500 μm .

b. The release profile of Rhodamine B loaded in microneedles with and without gold coating. Height of the microneedles, 1.2-mm; Thickness of the gold coating, 100 nm. All samples were immersed in 1X Dulbecco's PBS at 40 °C. **c.** Optical image of a wireless SOP. Scale bar: 5 mm. **d.** Top-down view of the wireless module used in the SOP, featuring the receiving coil (labeled as Rec. coil), the full-bridge rectifier and the regulator (circled in blue), the power amplifier (circled in green), the System-on-Chip (SoC) module (circled in red), and the ground (GND) & output channels. Both GND and output channels are connected to gold-coated MN arrays via conductive traces, which serve as a counter electrode and a working electrode, respectively. Scale bar: 5 mm. **e.** Measurement on power harvesting of the wireless SOP (without SoC module). Black, input AC signal (peak-to-peak 4 V, 39 MHz); Red, rectified received signal (3.4 V, stand deviation (s.d.) \sim 0.2 V); Blue, regulated output signal (2.6 V, s.d. \sim 0.1 V). **f.** The frequency matching characterization of the wireless power transfer. Black dots: the output voltage after rectification without regulation; Red dots: the output voltage after rectification and 2.5-V regulation. 10-V peak-to-peak input signal is used, and the working range from 32-40 MHz is determined at 36 MHz. The potential rectification achieves stable bias around 2.5 V as the electrical trigger. **g.** The circuit diagram of the energy harvesting module of SOP with correspondence to **e** and **d**. Inductive coil (Ind. coil) is not present in **c** and **d**. The power amplification and SoC module are abbreviated as Amp and SoC. CE stands for counter electrodes.

Figure 5. In vivo demonstration of SOP. **a.** Optical images of an intracranial microneedle (height, 3 mm) at various fabrication stages, 1) PLGA needle base; 2) loading melatonin (20 %); 3) coating with 150-nm gold. The drug-concentrated region is circled with a red frame. Scale bar: 1 mm. **b.** The measured force-displacement curve of a microneedle array (9 needles, 1.2 mm in length) during a fracture test. The first fracture point is circled by a red frame and the displacement in contact is labeled and measured as 1.26 mm. **c.** The schematic illustration of a mouse brain model indicating deployment location of SOP in the in vivo study. The red and white circles label the positions of the SOP microneedle and counter electrode (Pt wire), respectively; the blue and black circles label the positions of two separate recording electrodes (Pt wire). Scale bar: 5 mm. **d.** Optical images of the microneedle (3 mm in length) before test, after pulses trigger, and after square wave trigger, respectively. Scale bar: 1 mm. **e.** Measured square-wave triggers generated from the SOP microneedle at various distances to the microneedle. The trigger is delivered by the gold-coated microneedle (3-mm, 150-nm) as a 2.5-V 5-s square wave periodically. The two recording electrodes (Rec. 1 and Rec. 2) are around 5 mm away from the microneedle. **f.** Measured pulsatile triggers generated from the SOP microneedle at various distances to the microneedle. A pulse signal (50 mV, 1 Hz, 10 ms

in width) is applied by a microneedle electrode and recorded by a Pt wire at ~ 0.1, 2.0, and 5.0 mm, respectively. The signal for stimulation is attached in red. Scale bar: 5 s, 50 mV, for the x and y axis, respectively. **g.** Image of the freely moving mouse after SOP deployment. **h.** Immunohistochemical staining images of the recovery process after microneedle implantation. The combined images include Nissl bodies (neurotrace, green), astrocytes (glial fibrillary acidic protein (GFAP), red), activated microglia (Iba1, white) and nuclear DNA (4',6-diamidino-2-phenylindole (DAPI), blue). Scale bar: 100 μm .

SI Figure 16. The circuit diagram of SOP with a wireless power harvesting system and a remote-control module. A power harvesting and signal conditioning circuit is fabricated and soldered using the method of soft PCB fabrications. A power harvesting network formed by a PCB inductor coil and a surface mount capacitor is used to harvest energy from the alternating magnetic flux generated from the inductively coupled transmission coil. The power harvesting network is connected to a full-bridge rectifier with surface-mount diodes, enabling the conversion of the harvested alternative current (AC) obtained via the receiver coil into direct current (DC). Subsequently, the rectified DC is fed into a 3.3 V low-dropout (LDO) regulator. The coil, rectifier, and regulator constitute the wireless energy harvesting system, which is circled in a blue frame. After the regulator, the output signal goes through a power amplification system, which is circled in green. The amplified output serves as the power source for a Bluetooth-Low-Energy (BLE) System-on-Chip (SoC). A custom BLE service programmed in the SoC enables clinicians to wirelessly and remotely control the general-purpose input/output (GPIO) modules through Bluetooth using computers or smartphones. This scheme, in conjunction with the

programmed pulse width modulation (PWM) instance, onboard power amplifiers, and filtering circuits, facilitates programmable drug release by clinicians from the microneedle patch with high spatiotemporal resolution.

Modifications to the manuscript: Based on the reviewer's comments, we have made the following revisions:

On page 4 in the revised manuscript, we replaced **Figure 1a** with a new schematic illustration, including two parts: NFC module and MN patch. Correspondingly, we modified the figure caption of **Figure 1**.

On page 6 in the revised manuscript, we adjusted the legends in **Figure 2f** and removed the title of **Figure 2b-2c, 2e-2h**. In **Figure 2b-2c**, the x-axis labels were replaced by "before, transition, release" instead of "before, middle, after". They are consistent with **Figure 2a** now. Correspondingly, we revised the figure caption of **Figure 2**.

On page 10 in the revised manuscript, we reorganized the order of **Figure 3** in a more readable way. The titles of **Figure 3b, 3d-3f** were removed and described in the caption. Correspondingly, we revised the figure caption of **Figure 3**.

On page 12 in the revised manuscript, the title of **Figure 4b** was removed and described in the caption. Correspondingly, we revised the figure caption of **Figure 4**.

On page 14 in the revised manuscript, we reorganized the order of **Figure 5** in a more readable way. The titles of **Figure 5b, 5d-5f** were removed and described in the caption. Correspondingly, we revised the figure caption of **Figure 5**.

The font of the text was adjusted and unified according to the comment in all figures. We also changed the scale bars in every figure and labeled them in the caption instead of in the images.

Comment 2: "Authors should explain how the thickness of gold layer was tailored and measured. Also, EDS mapping should be utilized to further ensure the gold distribution."

Our response: We thank the referee for this comment. The gold layer is deposited on microneedles by sputter coating. This is a common method of physical vapor deposition that allows for omnidirectional deposition on mildly uneven surfaces at a specific rate (~0.2 nm/s). Therefore, the thickness of the gold coating is controlled by the deposition time. For example, the 150-nm gold film is introduced by a ~750-s deposition process. Regarding the uniformity, we conducted the cross-section SEM characterization of gold-coated microneedles. To further analyze the uniformity of the gold layer, we characterized the cross-section of an MN (150 nm gold coated, 1.5 mm) by SEM and a series of AFM

studies to reveal the surface roughness of gold deposition on a polymeric substrate. The SEM images of the MN cross-section show the detailed structure of the outer gold layer on the MN. In **SI Figure 10a-c**, we exposed the cross-section of MN with gold coating by freezing-breaking the MN. **SI Figure 10c** shows the cross-section of the gold layer (colored in orange), which is approximately estimated at 160 nm in width. Considering the existence of a diffusion layer (where gold atoms mix with polymer), the measurement error in thickness is acceptable. In **SI Figure 10d-f**, we prepared another MN (100 nm gold coated, 1.5 mm in MN height) with partial gold coating manually exfoliated. The thickness calculation based on perspective correction is around 95 nm, corresponding with the designed thickness. Both samples show good uniformity of the thickness of the gold layer. We conducted surface profilometry of the gold layer (150 nm, sputter coated) on a polymeric substrate to further characterize the film uniformity by sputter coating. Five different locations (5 μm in length) are presented in **SI Figure 11**. The mean average and root-mean-square (RMS) deviation are 1.74 and 2.32 nm, which is significantly lower than the film thickness. Besides, gold EDS mappings are provided in **SI Figure 9** to show the uniform distribution of gold on the MNs.

SI Figure 10. The SEM of Au film cross sections. **a. b. c.** The SEM images of an MN (150 nm gold coated, 1.5 mm) under various magnifications. **d. e. f.** The SEM images of a partially exfoliated gold-coated MN (100 nm gold coated, 1.5 mm) under various magnifications. The magnified areas are approximately labeled by red dash frames in **a** and **d**, and blue dash frames in **b** and **e**. Cross-sections of gold film are colored orange, gold film surfaces are colored yellow,

and the PLGA cross-section is colored light blue. Scale bars: 100 μm in **a** and **d**, 5 μm in **b** and **e**, 1 μm in **c** and **f**.

SI Figure 11. Surface profile of sputter-coated Au film on polymer substrate by AFM. Five different positions (labeled as Position 1-5) are characterized by AFM with roughness

calculated. Both 2D and 3D maps are provided for each position. RMS roughness (R_q) and arithmetic roughness (R_a) are attached to the 3D maps. Scale bar: 1 μm .

SI Figure 9. EDXS element mapping of oxygen (O), carbon (C), and gold (Au) from different stages of electrochemical crevice corrosion on two parts of the microneedle (the tip and the waist). The oxygen, carbon, and gold mappings are in yellow, green, and red, respectively. a. The high magnification SEM image, oxygen, and carbon mapping of the tip area of MNs from three stages: i. standby; ii. transitioning; iii.

releasing. **b.** The high magnification SEM image, oxygen, and carbon mapping of the waist area of MNs from three stages: **i.** standby; **ii.** transitioning; **iii.** releasing. Scale bar: 60 μm .

Modifications to the manuscript: Based on the reviewer's comments, we have made the following revisions:

On page 9 in the revised manuscript, we added "**Figure 2j** and **SI Figure 10** show the uniformity of the gold layer on MN prior to crevice corrosion (**SI Note 1**). We also use atomic force microscopy (AFM) to characterize the surface roughness of the gold layer deposited on polymer to illustrate the uniformity of deposition thickness by sputter coating, as shown in **SI Figure 11**."

On page 2 in the *Supplementary Information* section, we added **SI Note 1**, "To analyze the uniformity of the gold layer, we prepare two MN samples: 1) a 1.5 mm MN with 150 nm gold layer is broken in half after freezing (**SI Figure 10a-c**); 2) a 1.2 mm MN with 100 nm gold layer undergoes partial Au film exfoliation (**SI Figure 10d-f**). By SEM, the cross-section images of the MN reveal the detailed structure of the outer gold layer. **SI Figure 10c** shows the cross-section of the gold layer (colored orange), which is approximately estimated at 160 nm in width. Considering the existence of a diffusion layer (where gold atoms mix with polymer), the error in thickness is acceptable. Namely, **SI Figure 10f** shows the thickness of the exfoliated gold film, roughly calculated to be ~ 95 nm, after perspective correction. Both samples show good uniformity of the thickness of the gold layer, which provides good encapsulation performance. Furthermore, we utilize AFM to characterize the surface roughness of the coated gold layer on the polymer substrate, as presented in **SI Figure 10**. The average and root-mean-square (RMS) roughness is calculated to be less than 3 nm, which meets the requirement of thickness uniformity."

Figure 2j was added to **Figure 2** as a cross-section SEM of the gold layer on an MN. Correspondingly, we revised the figure caption of **Figure 2**.

On pages 14-15 in *Supplementary Information* section, we added **SI Figure 10** and **11**.

On page 13 in *Supplementary Information* section, we revised **SI Figure 9** by adding the EDXS mapping of the gold element. Correspondingly, we revised the figure caption of **SI Figure 9**.

Comment 3: "A schematic illustration about how the electrically controlled crevice corrosion of the SOP works should be provided."

Our response: We thank the referee for this comment. The mechanism of crevice corrosion is the key to the digital control of our SOP system, and it is of great importance to explain the reaction at the microscale level in detail. The SOP can be considered as a

two-electrode system, where the anode is the gold-coated microneedles. The cathode can be either gold-coated microneedles or other biocompatible conductors like platinum electrodes, to allow for the counter reaction. Although the biofluid environment is complicated, the main reactions can be considered as follows:

The existence of a buffer system in biofluids stabilizes the pH from sharp changes due to the cathode and anode reactions. In fact, the amount of gold on an MN array is estimated to be 290 μg , i.e., 1.47 μmol , there would be no observable changes in the pH of the biofluid. In addition, there can be other degradation mechanisms of gold film at the microscale level,¹⁻³ that some parts of gold are mechanically exfoliated due to crevices. These gold pieces are unstable and may quickly degrade into nanoparticles since the thickness is only ~ 150 nm. Here, we prepared another schematic illustration (**SI Figure 5**) to provide a better understanding of the electrode reaction. **SI Figure 5** briefly illustrates the two-electrode system of our SOP. With positive potential applied to the anode, the gold layer exposed in the biofluid will go through typical anodic oxidation, resulting in Au_2O_3 particles. **SI Figure 5b-c** provides a rough illustration of this reaction at the atomic level: **b**. the gold surface without potential applied; **c**. the gold surface that undergoes crevice corrosion and the resulting oxides.

SI Figure 5. Schematic illustration of the crevice corrosion of gold in biofluids. a. Schematic illustration of anode and cathode half-reactions of the two-electrode system in SOP. **b. c.** The microscopic illustration of the anodic oxidation reaction of gold ([111] facet). **b.** Standby stage before crevice corrosion (without triggering potential applied). **c.**

Transitioning stage during crevice corrosion (with triggering potential applied). Au, O, and H atoms are in golden, white, and red colors, respectively.

Modifications to the manuscript: Based on the reviewer's comments, we have made the following revisions:

On page 8 in the revised manuscript, we added: “**SI Figure 5** provides a schematic illustration of the two-electrode system of SOP (**SI Figure 5a**) and the atomic scale illustration of the anodic oxidation (**SI Figure 5b-c**) on Au (111) facet in a neutral environment, with reaction products, Au₂O₃ in the form of nanoparticles.”

On page 9 in *Supplementary Information* section, we added **SI Figure 5**.

We added the following references in the revised manuscript:

1. Santini, J. T., Cima, M. J. & Langer, R. A controlled-release microchip. *Nature* **397**, 335–338 (1999).
2. Mirvakili, S. M. & Langer, R. Wireless on-demand drug delivery. *Nat. Electron.* **4**, 464–477 (2021).
3. Li, Y., Shawgo, R. S., Langer, R. & Cima, M. J. Mechanical testing of gold membranes on a MEMS device for drug delivery. *2nd Annu. Int. IEEE-EMBS Spec. Top. Conf. Microtechnologies Med. Biol. - Proc.* 390–393 (2002)

Comment 4: “Safety assessment of SOP should be carried out to obtain a more comprehensive understanding of the potential side effects. This additional research would enhance the credibility and reliability of the findings.”

Our response: We thank the referee for this comment. We conducted the following experiments to obtain a more comprehensive assessment of the safety of intracranial MNs, as the drug-delivery interface of SOP.

First, we conduct a series of evaluations on the inflammation and neural damage among four groups (n= 6-8 per group) of MNs used in the intracranial implantation of mice, including one implanted with bare MN (MN), one with gold-coated MN (Au-MN), one with melatonin-loaded MN (Mel-MN), and one with melatonin loaded MN with gold coating (Au-Mel-MN). From the typical images of MN implanting sites, the number of astrocytes and microglia in the region of interest (ROI) is significantly lower in the Au-MN or Au-Mel-MN group compared to that in the bare MN group, which indicates that gold-coated MNs have less damage to neural tissues. Meanwhile, fluorescence density of the markers has no difference between bare and gold-coated MNs, which indicates Au encapsulation on MNs does not affect the expressing strength of the GFAP or IBA signals.

To better assess the safety and cytotoxicity, we conducted Ca²⁺ imaging to analyze the effect of MN implantation on the level of neural activity. As illustrated in **SI Figure 22**, we

observe no significant difference in Ca^{2+} activities between the control (sham operation only), MN, and Au-MN groups. This indicates that the implanted MN does not have a negative influence on neural activity, showing no significant cytotoxicity to the peripheral neural tissues.

SI Figure 21. Immunohistochemical analysis to validate biocompatibility of SOP. The study compared four groups of MNs, including bare MN (MN), gold-coated MN (Au-

MN), melatonin-loaded MN (Mel-MN), and melatonin-loaded MN with gold-coating (Au-Mel-MN). **a.** Representative confocal images of 40- μm horizontal cortical slices after intracranial implantation of the MNs in mice. The images show cross-sectional views of the implantation site with immunohistochemical staining for 1) astrocytes (glial fibrillary acidic protein (GFAP), green); 2) activated microglia (Iba1, white); 3) Neuron-specific enolase (NSE, red) and 4) DNA (4',6-diamidino-2-phenylindole (DAPI), blue), and overall lesions from bioresorbable MNs. **b.** Measured the number of astrocytes and microglia in the region of interest (ROI). **c.** Average fluorescence intensity of astrocytes and microglia in the region of interest (ROI). ($n = 6-8$ in each group). $*p < 0.05$. One-way ANOVA followed by PLSD post hoc test. Scale bar: 100 μm .

SI Figure 22. Recorded Ca^{2+} activities of the prefrontal cortex (PFC) after sham operation and implantation of the bare or the gold-coated MNs. **a.** The schematic illustration and fluorescence images that shows fiber photometry recording after intracranial implantation of MNs in mice. Scale bar: 50 μm . The positions of the optical fiber and MN are labeled. GCaMP6f, Td, and DAPI are in green, red, and blue. **b.** Typical GCaMP6f traces (5 mins) of the sham group (Control, upper), the group with bare MNs (MN, middle), and the group with the gold-coated MNs (Au-MN, bottom) from the PFC. **c.** The cumulative activity, frequency, and average peak value of DF/F of the PFC calcium signals. ($n = 6$ in each group)

Modifications to the manuscript: Based on the reviewer's comments, we have made the following revisions:

On page 16 in the revised manuscript, we added “**SI Figure 20** shows an in vivo study to validate the safety and biocompatibility of SOP in intracranial deployment. Here, the study compares four groups of MNs (bare MN (MN), gold-coated MN (Au-MN), melatonin-loaded MN (Mel-MN), and melatonin-loaded MN with gold coating (Au-Mel-MN)) in their effects on the inflammation and degree of neural damage after one-month implantation. **SI Figure 21a** shows typical images of MN implanting sites with GFAP/IBA/NSE/DAPI multiple staining. As a result, the number of astrocytes and microglia in the region of interest (ROI) is significantly lower in the Au-MN or Au-Mel-MN group compared to the bare MN group, which indicates that gold-coated MNs have less damage to neural tissues and cause less inflammation compared to bare PLGA MNs. In addition, the fluorescence density of the markers has no difference between bare and gold-coated MNs, which indicates that the Au encapsulation on MNs does not affect the expressing strength of the GFAP or IBA signals (**SI Figure 21c**).

Furthermore, we used a fiber photometry system to investigate the neural activity in the medial prefrontal cortex (mPFC) after the implantation of MNs (bare MN, Au-MN, control (no MN), respectively, **SI Figure 22**). As shown in **SI Figure 22c**, there is no observable difference in Ca^{2+} dynamics of mPFC excitatory neurons between the control group, bare MN, and Au-MN groups, which indicates the MNs have no significant damage to the brain tissue and introduce little influence on the neuronal activity.”

On pages 26-27 in *Supplementary Information* section, we added **SI Figures 21** and **22**.

On pages 2-3 in *Supplementary Information* section, we added **SI Note 2**,

“Immunohistochemical analysis (additional)

Brain sections were then incubated for ~16 h at 4 °C in a blocking buffer containing goat anti-GFAP of Santa Cruz Biotechnology (1:1000), rabbit anti-Iba1 of Fujifilm Wako (1:500), and Chichen anti-Neuron Specific Enolase (NSE Millipore AB9698, 1:100).

All data values were presented as the mean \pm standard deviation (SD). For analyzing fluorescence density and mean of peak value in the region of interest (ROI), we define the ROI as the areas within 200 μ m lateral to the edge of the probes. One-way ANOVA was used to compare the fluorescence density and mean of peak value between different experimental groups.

As is well known, GFAP and IBA-1, representing astrocytes and microglia, respectively, reflect the inflammation in the brain tissues^{1,2}. Neuron-specific enolase (NSE), an acidic protease unique to neurons and neuroendocrine cells and a sensitive indicator for assessing the severity of nerve cell damage and prognosis, is widely used to reflect neural injuries and pathological processing³.

Stereotaxic surgery (additional)

For in vivo photometry recordings, mice were unilaterally injected with 250 nl of AAV5-CaMKII α -CaMP6f mixed with 5 nl AAV5-CaMKII-tdTomato into the mPFC at these coordinates AP: -1.5 mm, ML: +0.3 mm, DV: -2.1 mm. Optical fibers (Newdoon Inc, O.D.: 1.25 mm, core: 200 μ m, NA: 0.37) were implanted 0.2 mm above the mPFC (AP: -1.5 mm, ML: +1.4 mm, DV: -1.9 mm). At the same time, MN probes were implanted ipsilaterally into the medial prefrontal cortex at 30 degrees lateral to the optical fiber.

Fiber photometry system

The multi-fiber photometry system was used as previously described^{4 5}. Briefly, the system consisted of a 488 nm excitation laser, a fluorescence cube, and a spectrometer. The 488 nm laser beams first launched into the fluorescence cube and then into the optical fibers. The GCaMP and tdTomato emission fluorescence collected from the fiber probe traveled back to the spectrometer. Only animals with strong GCaMP and tdTomato expression were included in the study (**SI Figure 22a**). Spectral data was acquired by OceanView software (Ocean Optics, Inc) at 10 Hz and was synchronized to a 20 Hz video recording system to acquire the animal behavior.

The in vivo recordings were carried out in an open-top home cage (21.6 “ 17.8 “ 12.7 cm) in the 30 Lux red light environment. Laser power was adjusted to a final optical fiber output of 30 mW. Photometry data were exported to MATLAB R2014b for analysis. Coefficients of GCaMP6f and tdTomato were unmixed by a customized script by fitting spectrum signals to standard emission curves. GcaMP6f signals were normalized by tdTomato signals for motion correction. (**SI Figure 22a**). A 0.1 Hz high-pass filter corrected the fluorescence bleaching. Photometry signals ($\Delta F/F$) were derived by calculating $(F-F_0)/F_0$, where F_0 is the median of the fluorescence signal (**SI Figure 22b**). For the home cage analysis, we recorded data for 5 min per mouse and calculated the $\Delta F/F$ to further analyze the correlation of SuM and DG signals in raw data (**SI Figure 22c**). Cumulative activity = $\Delta F/F \times \text{Time}$. The average of the peak of $\Delta F/F$ above 2SD during 5 mins in the home cage was calculated.”

We added the following references in the *Supplementary Information* section, corresponding to **SI Note 2**:

1. Giovannoni, F. & Quintana, F. J. The Role of Astrocytes in CNS Inflammation. *Trends Immunol.* **41**, 805–819 (2020).
2. Muzio, L., Viotti, A. & Martino, G. Microglia in Neuroinflammation and Neurodegeneration. *Front. Neurosci.* **15**, 742065 (2021).
3. Thelin, E. P. *et al.* Utility of neuron-specific enolase in traumatic brain injury; relations to S100B levels, outcome, and extracranial injury severity. *Crit. Care* **20**, 285 (2016).
4. Meng, C. *et al.* Spectrally Resolved Fiber Photometry for Multi-component Analysis of Brain Circuits. *Neuron* **98**, 707-717.e4 (2018).
5. Li, Y. *et al.* Supramammillary nucleus synchronizes with dentate gyrus to regulate

spatial memory retrieval through glutamate release. *Elife* **9**, e53129 (2020).

Comment 5: “The reference part was organized in a poor way. Some lacked the author's full surname and some lacked the specific page numbers or article numbers.”

Our response: We thank the referee for this comment. We have carefully reviewed and revised the reference section to ensure consistency. Regarding references, we go through each of them to investigate possible errors. Some author names are corrected (Reference No. 2, 5, 48, 79, 80, 83), as the full surname of them was not provided in a proper way. Some page numbers or article numbers are added (Reference No. 1, 6, 11, 13, 16, 17, 20, 22, 24, 31, 32, 33, 35, 38, 39, 43, 44, 45, 47, 51, 52, 63, 66, 68, 72, 76, 78, 88, 90). Some references are corrected in format (Reference No. 30, 42, 54, 84). In addition, we replaced reference No. 73 with the following one, as it is more specified to the data we mentioned.

Modifications to the manuscript: Based on the reviewer’s comments, we have made the following revisions:

We corrected the references in the revised manuscript, here are some examples.

2. Teo, B. M. & Seah, B. C.-Q. Recent advances in ultrasound-based transdermal drug delivery. *Int. J. Nanomedicine* **13**, 7749–7763 (2018).

17. Hsu, W. L. *et al.* On-skin glucose-biosensing and on-demand insulin-zinc hexamers delivery using microneedles for syringe-free diabetes management. *Chem. Eng. J.* **398**, 125536 (2020).

24. Li, W. *et al.* Long-acting reversible contraception by effervescent microneedle patch. *Sci. Adv.* **5**, eaaw8145 (2019).

30. Yin, Y. *et al.* Separable Microneedle Patch to Protect and Deliver DNA Nanovaccines against COVID-19. *ACS Nano* **15**, 14347–14359 (2021).

54. Tait, W. S. Electrochemical corrosion basics. in *Handbook of Environmental Degradation Of Materials: Third Edition* 97–115 (2018).

73. Neal, N. C. & Burke, F. D. High-pressure injection injuries. *Injury* **22**, 467–470 (1991).

Comment 6: “The English writing needs improving as there are many grammatical errors over the manuscript.”

Our response: We thank the referee for this comment. We carefully checked the entire manuscript including figures and supporting information and corrected those errors. The

main text, figures, figure captions, supporting information, and other sections have been carefully reviewed. A major part of the revision is focused on correcting grammatical errors, as kindly mentioned in the comment. We also unify the expression to our best. Overall, we highly appreciate the referee for providing this comment, as an opportunity for us to improve the manuscript.

Modifications to the manuscript: Based on the reviewer's comments, we have made the following revisions:

We unify the expression of term. For example, "gold", except for some special cases like "Au-MN", are unified instead of using "Au".

We check the abbreviations and make sure they are at least mentioned in the full term when they first appear. For example, "MN" for microneedle.

We check the singular or plural form of nouns and verbs. They are now consistent with what they're used to describe. For example, to describe a microneedle array, "microneedles" and "MNs" are used instead of "microneedle" and "MN".

We make sure that we use present tense and active voice in most cases in the main text. For example, "**SI Figure 5** provides a schematic illustration of the two-electrode system of SOP (**SI Figure 5a**) and the atomic scale illustration of the anodic oxidation (**SI Figure 5b-c**) on Au (111) facet in a neutral environment, with reaction products, Au₂O₃ in the form of nanoparticles."

We reorganize the expressions in the manuscript to avoid redundancy and improve conciseness. For example, we use "drug delivery" instead of "the delivery of drug".

Responses to comments of Referee #2

Summary Comment: “This study reports a novel transdermal drug delivery system called the "spatiotemporal on-demand patch (SOP)". The system combines drug-loaded microneedles with biocompatible metal films to achieve digitally controlled drug release with high precision in both space and time. Some concerns should be addressed, as follows:”

Our response: We thank the referee for these positive comments and for these helpful suggestions for revision. We carefully addressed the issues, as listed below, and revised our manuscript accordingly.

Modifications to the manuscript: None.

Comment 1: “The active release behavior of the SOP system can be effectively initiated by opening the Au film gate by corrosion-induced crevices via electrical triggers. However, in the manuscript, the authors do not give a detailed explanation of the reactions that occur between the anode and cathode. Please provide a more detailed explanation of how the drug delivery system works and how the mechanism of electrically controlled drug release is implemented.”

Our response: We thank the referee for this comment. The mechanism of crevice corrosion is the key to the digital control of our SOP system, and it is of great importance to explain the reaction at the microscale level in detail. The SOP can be considered as a two-electrode system, where the anode is the gold-coated microneedles. The cathode can be either gold-coated microneedles or other biocompatible conductors like platinum electrodes, to allow for the counter reaction. Although the biofluid environment is complicated, the main reactions can be considered as follows:

The existence of a buffer system in biofluids stabilizes the pH from sharp changes due to the cathode and anode reactions. In fact, the amount of gold on an MN array is estimated to be 290 μg , i.e., 1.47 μmol , there would be no observable changes in pH of the biofluid. In addition, there could be other degradation mechanisms of gold film at the microscale level,¹⁻³ that some parts of gold are mechanically exfoliated due to crevices. These gold pieces are not stable and may quickly degrade into nanoparticles since the thickness is only ~ 150 nm. Here, we prepared another schematic illustration (**SI Figure 5**) to provide a better understanding of the electrode reaction. **SI Figure 5** briefly illustrates the two-electrode system of our SOP. With positive potential applied to the anode, the gold layer exposed in the biofluid will go through typical anodic oxidation, resulting in Au_2O_3 particles. **SI Figure 5b-c** provide a rough illustration of this reaction at the atomic level: **b.** the gold

surface without potential applied; **c.** the gold surface that undergoes crevice corrosion and the resulting oxides.

SI Figure 5. Schematic illustration of the crevice corrosion of gold in biofluids. a. Schematic illustration of anode and cathode half-reactions of the two-electrode system in SOP. **b. c.** The microscopic illustration of the anodic oxidation reaction of gold ([111] facet). **b.** Standby stage before crevice corrosion (without triggering potential applied). **c.** Transitioning stage during crevice corrosion (with triggering potential applied). Au, O, and H atoms are in golden, white, and red, respectively.

Modifications to the manuscript: Based on the reviewer’s comments, we have made the following revisions:

On page 8 in the revised manuscript, we added “**SI Figure 5** provides a schematic illustration of the two-electrode system of SOP (**SI Figure 5a**) and the atomic scale illustration of the anodic oxidation (**SI Figure 5b, c**) on Au (111) facet in a neutral environment, with reaction products, Au₂O₃ in the form of nanoparticles.”

On page 9 in *Supplementary Information* section, we added **SI Figure 5**.

We added the following references in the revised manuscript:

1. Santini, J. T., Cima, M. J. & Langer, R. A controlled-release microchip. *Nature* **397**, 335–338 (1999).
2. Mirvakili, S. M. & Langer, R. Wireless on-demand drug delivery. *Nat. Electron.* **4**, 464–477 (2021).

3. Li, Y., Shawgo, R. S., Langer, R. & Cima, M. J. Mechanical testing of gold membranes on a MEMS device for drug delivery. *2nd Annu. Int. IEEE-EMBS Spec. Top. Conf. Microtechnologies Med. Biol. - Proc.* 390–393 (2002)

Comment 2: “Line 75-77, the authors mentioned that a typically used metal gate for forming electronic implants that enable on-demand drug delivery is a gold layer with a thickness of approximately 300 nm. Why does the authors choose a 150 nm thick Au layer in this work?”

Our response: We thank the referee for this comment. Yes, as described in the manuscript, previously reported work mostly use a thicker gold layer (300 nm). In their designs, the gold layer is a suspended diaphragm on the opening of the drug reservoir. Therefore, mechanical robustness achieved by using a relatively thicker gold layer is an important factor in these designs.¹⁻⁴ The metal film needs to be thick to provide enough stability against cracking and collision. However, in our design, the gold layer for drug encapsulation is coated and supported on the surface of microneedles. Thus, it is already supported and stabilized by the microneedle structure, thus allowing a thinner form factor compared with those of the previous works.

SI Figure 1. The comparison between microneedles and drug reservoirs with metallic gates. a. Schematic illustration of the drug-loaded microneedles with a layer of gold coating. **b.** The microscale illustration of the gold-PLGA interface in **a**. **c.** Schematic illustration of the drug reservoir with a liquid payload and a gate of gold membrane. **d.** The microscale illustration of the interface between the suspended gold membrane and liquid drug solution in **c**.

Modifications to the manuscript: Based on the reviewer's comments, we have made the following revisions:

On page 5 in the revised manuscript, we added "The gold encapsulation layer is supported on the surface of solid MNs, which enables sufficient stability with a thinner thickness (150 nm) compared with those used in the reservoir designs (thickness of gold layer more than 300 nm) as reported previously (as illustrated in **SI Figure 1**). Thus, the thickness to realize effective drug encapsulation can be smaller."

On page 4 in *Supplementary Information* section, we added **SI Figure 1**.

We added the following references in the revised manuscript:

1. Li, Y., Shawgo, R. S., Langer, R. & Cima, M. J. Mechanical testing of gold membranes on a MEMS device for drug delivery. *2nd Annu. Int. IEEE-EMBS Spec. Top. Conf. Microtechnologies Med. Biol. - Proc.* 390–393 (2002)
2. Mirvakili, S. M. & Langer, R. Wireless on-demand drug delivery. *Nat. Electron.* **4**, 464–477 (2021).
3. Koo, J. *et al.* Wirelessly controlled, bioresorbable drug delivery device with active valves that exploit electrochemically triggered crevice corrosion. *Sci. Adv.* **6**, eabb1093 (2020).
4. Nadeau, P. *et al.* Prolonged energy harvesting for ingestible devices. *Nat. Biomed. Eng.* **1**, 0022 (2017).

Comment 3: "The thickness of the gold layer has a significant effect on drug release. How the authors precisely control the thickness of the Au coating in this work? And what about the uniformity of the Au coating? Could you please provide a cross-section picture of the Au coated microneedles to demonstrate the gold coating thickness on the surface of PLGA microneedles? What about the bonding force between the gold coating and the surface of PLGA microneedles? And is there a risk of the gold layer falling off during the insertion of the microneedles into the skin?"

Our response: We thank the referee for this comment. The gold layer is deposited on microneedles by sputter coating. This is a common method of physical vapor deposition that allows for omnidirectional deposition on mildly uneven surfaces at a certain rate (~ 0.2 nm/s). Therefore, the thickness of the gold coating is controlled by the time of deposition. For example, the 150-nm gold film is introduced by a ~ 750-s deposition process. Regarding the uniformity, we conducted the cross-section SEM characterization of gold-coated microneedles.

In order to further analyze the uniformity of gold layer, we characterized the cross-section of an MN (150 nm gold coated, 1.5 mm) by SEM, and a series of AFM studies to reveal the surface roughness of gold deposition on a polymeric substrate. The SEM images of

MN cross-section reveal the detailed structure of the outer gold layer on the MN. In **SI Figure 10a-c**, we exposed the cross-section of MN with gold coating by freezing-breaking the MN. **SI Figure 10c** shows the cross-section of the gold layer (colored in orange), which is approximately estimated at 160 nm in width. Considering the existence of a diffusion layer (where gold atoms mix with polymer), the measurement error in thickness is acceptable. In **SI Figure 10d-f**, we prepared another MN (100 nm gold coated, 1.5 mm in MN height) with partial gold coating manually exfoliated. The thickness calculation based on perspective correction is around 95 nm, which corresponds with the designed thickness. Both samples show good uniformity of the thickness of gold layer.

To further characterize the film uniformity by sputter coating, we conducted surface profilometry of gold layer (150 nm, sputter coated) on a polymeric substrate. Five different locations (5 μm in length) are presented in **SI Figure 11**. The mean average and root-mean-square (RMS) deviation are 1.74 and 2.32 nm, respectively, which is significantly lower than film thickness.

Apart from the uniformity of the gold coating, the bonding force between gold layer and PLGA MN is a key point to be considered. In the previous manuscript, the bonding stability is characterized by PBS soaking and agar penetration experiments, as shown in **Figure 3** and **SI Figure 15**. Here, we present another experiment that further validates the stability of gold-PLGA bonding. **SI Figure 14** shows a series of penetration experiments using our MN patch (3*3 array, 150 nm gold coated, 1.2 mm) on chicken thigh tissue. Within 20 times contact and penetration test, there are no significant changes in the gold film, as shown in the optical images.

SI Figure 10. The SEM of Au film cross sections. a. b. c. The SEM images of an MN (150 nm gold coated, 1.5 mm) under various magnifications. **d. e. f.** The SEM images of a partially exfoliated gold-coated MN (100 nm gold coated, 1.5 mm) under various magnifications. The magnified areas are approximately labeled by red dash frames in **a** and **d**, and blue dash frames in **b** and **e**. Cross-sections of gold film are colored orange, gold film surfaces are colored yellow, and the PLGA cross-section is colored light blue. Scale bars: 100 μm in **a** and **d**, 5 μm in **b** and **e**, 1 μm in **c** and **f**.

SI Figure 11. Surface profile of sputter-coated gold film on polymer substrate by AFM. Five different positions (labeled as Position 1-5) are characterized by AFM with roughness calculated. Both 2D and 3D maps are provided for each position. RMS

roughness (R_q) and arithmetic roughness (R_a) are attached to the 3D maps. Scale bar: 1 μm .

SI Figure 14. Optical images of gold-coated MN under penetration test. A 3*3 MN array (150 nm gold coated, 1.2 mm) is used in the penetration test of chicken thigh tissue. **a.** MN array before penetration. **b.** MN array after one penetration. **c.** MN array after 2 penetrations. **d.** MN array after 5 penetrations. **e.** MN array after 10 penetrations. **f.** MN array after 20 penetrations. **g.** The MN array facing upward on a chicken thigh. **h.** The MN array in contact with chicken thigh. Scale bars: 1 mm.

Modifications to the manuscript: Based on the reviewer’s comments, we have made the following revisions:

On page 9 in the revised manuscript, we added “**Figure 2j** and **SI Figure 10** show the uniformity of the gold layer on MN prior to crevice corrosion (**SI Note 1**). We also use

atomic force microscopy (AFM) to characterize the surface roughness of gold layer deposited on polymer to illustrate the uniformity of deposition thickness by sputter coating, as shown in **SI Figure 11.**”

On page 2 in *Supplementary Information* section, we added **SI Note 1**, “To analyze the uniformity of the gold layer, we prepare two MN samples: 1) a 1.5 mm MN with 150 nm gold layer is broken in half after freezing (**SI Figure 10a-c**); 2) a 1.2 mm MN with 100 nm gold layer undergoes partial Au film exfoliation (**SI Figure 10d-f**). By SEM, the cross-section images of the MN reveal the detailed structure of the outer gold layer. **SI Figure 10c** shows the cross-section of the gold layer (colored orange), which is approximately estimated at 160 nm in width. Considering the existence of a diffusion layer (where gold atoms mix with polymer), the error in thickness is acceptable. Namely, **SI Figure 10f** shows the thickness of the exfoliated gold film, roughly calculated to be ~ 95 nm, after perspective correction. Both samples show good uniformity of the thickness of the gold layer, which provides good encapsulation performance. Furthermore, we utilize AFM to characterize the surface roughness of the coated gold layer on the polymer substrate, as presented in **SI Figure 10**. The average and root-mean-square (RMS) roughness is calculated to be less than 3 nm, which meets the requirement of thickness uniformity.”

On page 11 in the revised manuscript, we added “**SI Figure 14**, **SI Figure 15e**, together with **Figure 3a**, show the stability of gold film on MNs against soaking in biofluids and mechanical friction in tissues. This indicates that the bonding between gold and PLGA is strong enough to meet the need for implantation, which corresponds with previous research on Au-polymer adhesion.⁵⁹⁻⁶¹”

On page 15 in the revised manuscript, we added “Another concern is the stability of the gold layer during implantation. **SI Figure 15e** shows the PLGA MN (150 nm gold coated, 1.2 mm) before and after the penetration test. No significant changes can be observed based on the comparison of optical images. **Figure 3a** also proves the stability of the gold layer during the soaking test. In addition, we use the MN array (150 nm gold coated, 1.2 mm) to penetrate the tissue of the chicken thigh for multiple cycles, as presented in **SI Figure 14**. The gold layer remains stable after 20 cycles of penetration.”

Figure 2j was added to **Figure 2** as a cross-section SEM of the gold layer on an MN. Correspondingly, we revised the figure caption of **Figure 2**.

On pages 14, 15, and 18 in *Supplementary Information* section, we added **SI Figure 10**, **11**, and **14**.

On page 13 in *Supplementary Information* section, we revised **SI Figure 9** by adding the EDXS mapping of gold element. Correspondingly, we revised the figure caption of **SI Figure 9**.

We added the following references in the revised manuscript:

1. Schonhorn, H., Roberts, R. F. & Hobbins, N. D. Bonding of polyethylene to gold. *J. Adhes.* **36**, 151–159 (1991).

2. Nordström, M. et al. Investigation of the bond strength between the photo-sensitive polymer SU-8 and gold. *Microelectron. Eng.* **78–79**, 152–157 (2005).

3. Takakuwa, M. et al. Direct gold bonding for flexible integrated electronics. *Sci. Adv.* **7**, eabl6228 (2021).

Comment 4: “Figure 3b suggested that the MN patch with gold coating showed excellent protection of encapsulated drugs from releasing. And the authors mentioned that “The average release rate of Rhodamine B in an hour is calculated as ~ 343 ng/min” (line 314 - 315). How is this result obtained from Figure 3b? In fact, there is a lack of data supporting the drug cumulative release kinetics of both PLGA microneedles and Au-coated microneedles. Please provide the cumulative drug release amount data of both W/ Au and W/O group at different time points based on Figure 3b.”

Our response: We appreciate the referee for this question. After reviewing the result, we found there is a minor error in the calculation. The error is fixed and will not affect our conclusion. **SI Figure 13** is also updated with the correct results.

In this experiment, we first obtained the calibration curve of Rhodamine B UV-Vis absorption in water, as demonstrated in **SI Figure 13b, c**. A linear regression curve is obtained together, as below:

$$Abs = -0.02998 + 0.09518 * C(RhB)$$

In this equation, *Abs* is the peak absorption (a.u.) at 556 nm, *C(RhB)* is the concentration ($\mu\text{mol/L}$) of Rhodamine B. The volume of solution in this experiment is controlled at 3 mL. Therefore, the molar amount of cumulative release (W/O group) at 60 min is calculated as follows:

$$n(RhB) = \frac{Abs_{60 \text{ min}} + 0.02998}{0.09518} * 0.003 \text{ L} = 0.052 \mu\text{mol}$$

Since the molar weight of Rhodamine B is 479.02 g/mol, the average drug release rate of Rhodamine B in 60 min is calculated as below:

$$Rate = 479.02 \text{ g/mol} * 0.052 \mu\text{mol} / (60 \text{ min}) = 415 \text{ ng/min}$$

Here we provide a table of the cumulative drug release amount data of both W/ Au and W/O Au groups at different time points:

Time / min	n(RhB) / μmol – W/O Au	n(RhB) / μmol – W/ Au
0	0.000610843	0.000610843

1	0.013033831	0.00059067
2	0.021904602	0.000601072
3	0.024912166	0.000608636
4	0.026847132	0.000607691
5	0.028640891	0.000619353
6	0.03008258	0.00067262
7	0.031771065	0.000660328
8	0.032870141	0.000650557
9	0.034568397	0.000657176
10	0.035853436	0.000967703
15	0.039121034	0.000699412
20	0.040754991	0.000747321
25	0.043185438	0.000677979
30	0.046155495	0.000704455
60	0.052038874	0.001282517

SI Table 1. The accumulative release amount of Rhodamine B (RhB). This table corresponds to the RhB release experiment described in **Figure 3b** and **SI Figure 13**. Both groups, microneedle array without and with gold encapsulation (W/O Au and W/ Au) are calculated for every recorded time point during the 60-min release.

SI Figure 13. Characterization of dye release from microneedles (0.3% Rhodamine B loaded, 1.2-mm). **a.** Optical images of an MN array undergoing dye release from 0 to 60 minutes in 45 °C 1X DPBS. Scale bar: 500 μm . **b.** The UV-Vis spectroscopy (300-800 nm) of Rhodamine B standard solutions (Sample 1-6). **c.** The calibration curve of Rhodamine B from standard solutions in **b.** **d.** Measured UV-Vis spectra of the environment solution corresponding to **a.** **e.** The absorbance versus time of environment solution from the dye release experiment.

Modifications to the manuscript: Based on the reviewer’s comments, we have made the following revisions:

On page 11 in the revised manuscript, we added “The calculation of the accumulative release amount is based on a calibration curve of Rhodamine B solutions and Beer-Lambert Law. Detailed explanation is provided in **SI Figure 13** and **SI Table 1.**”

On page 36 in *Supplementary Information* section, we added **SI Table 1.**

On page 17 in *Supplementary Information* section, we adjusted **SI Figure 13e-h.**

Comment 5: “Since acetone solvent is used in the preparation of microneedles, it is crucial to take into account the potential safety concerns arising from any residue left by the solvent. Therefore, conducting a cytotoxicity test becomes essential in order to assess the biocompatibility and safety of the microneedles. Please supplement the cytotoxicity test of the MNs system.”

Our response: We thank the referee for this comment. We conducted the following experiments to obtain a more comprehensive assessment of the safety of intracranial MNs, as the drug-delivery interface of SOP.

First, we conducted a series of evaluations on the inflammation and neural damage among four groups of MNs used in the intracranial implantation of mice, including one implanted with bare MN (MN), one with gold-coated MN (Au-MN), one with melatonin-loaded MN (Mel-MN), and one with melatonin loaded MN with gold coating (Au-Mel-MN). From the typical images of MN implanting sites, the number of astrocytes and microglia in the region of interest (ROI) is significantly lower in the Au-MN or Au-Mel-MN group compared to that in the bare MN group, which indicates that gold-coated MNs have less damage to neural tissues. Meanwhile, fluorescence density of the markers has no difference between bare and gold-coated MNs, which indicates gold encapsulation on MNs does not affect the expressing strength of the GFAP or IBA signals.

To better assess the safety and cytotoxicity, we conducted Ca^{2+} imaging to analyze the effect of MN implantation on the level of neural activity. As illustrated in **SI Figure 22**, we observe no significant difference in Ca^{2+} activities between the control (sham operation only), MN, and Au-MN groups. This indicates that the implanted MN does not have a negative influence on neural activity, showing no significant cytotoxicity to the peripheral neural tissues.

SI Figure 21. Immunohistochemical analysis to validate biocompatibility of SOP. The study compared four groups of MNs, including bare MN (MN), gold-coated MN (Au-MN), melatonin-loaded MN (Mel-MN), and melatonin-loaded MN with gold-coating (Au-Mel-MN).

Mel-MN). **a.** Representative confocal images of 40- μm horizontal cortical slices after intracranial implantation of the MNs in mice. The images show cross-sectional views of the implantation site with immunohistochemical staining for 1) astrocytes (glial fibrillary acidic protein (GFAP), green); 2) activated microglia (Iba1, white); 3) Neuron-specific enolase (NSE, red) and 4) DNA (4',6-diamidino-2-phenylindole (DAPI), blue), and overall lesions from bioresorbable MNs. **b.** Measured the number of astrocytes and microglia in the region of interest (ROI). **c.** Average fluorescence intensity of astrocytes and microglia in the region of interest (ROI). ($n = 6-8$ in each group). $*p < 0.05$. One-way ANOVA followed by PLSD post hoc test. Scale bar: 100 μm .

SI Figure 22. Recorded Ca^{2+} activities of the prefrontal cortex (PFC) after sham operation and implantation of the bare or the gold-coated MNs. **a.** The schematic illustration and fluorescence images that shows fiber photometry recording after intracranial implantation of MNs in mice. Scale bar: 50 μm . The positions of the optical fiber and MN are labeled. GCaMP6f, Td, and DAPI are in green, red, and blue. **b.** Typical GCaMP6f traces (5 mins) of the sham group (Control, upper), the group with bare MNs (MN, middle), and the group with the gold-coated MNs (Au-MN, bottom) from the PFC. **c.** The cumulative activity, frequency, and average peak value of DF/F of the PFC calcium signals. ($n = 6$ in each group)

Modifications to the manuscript: Based on the reviewer's comments, we have made the following revisions:

On page 16 in the revised manuscript, we added “**SI Figure 20** shows an in vivo study to validate the safety and biocompatibility of SOP in intracranial deployment. Here, the study compares four groups of MNs (bare MN (MN), gold-coated MN (Au-MN), melatonin-loaded MN (Mel-MN), and melatonin-loaded MN with gold coating (Au-Mel-MN)) in their effects on the inflammation and degree of neural damage after one-month implantation. **SI Figure 21a** shows typical images of MN implanting sites with GFAP/IBA/NSE/DAPI multiple staining. As a result, the number of astrocytes and microglia in the region of interest (ROI) is significantly lower in the Au-MN or Au-Mel-MN group compared to the bare MN group, which indicates that gold-coated MNs have less damage to neural tissues and cause less inflammation compared to bare PLGA MNs. In addition, the fluorescence density of the markers has no difference between bare and gold-coated MNs, which indicates that the Au encapsulation on MNs does not affect the expressing strength of the GFAP or IBA signals (**SI Figure 21c**).

Furthermore, we used a fiber photometry system to investigate the neural activity in the medial prefrontal cortex (mPFC) after the implantation of MNs (bare MN, Au-MN, control (no MN), respectively, **SI Figure 22**). As shown in **SI Figure 22c**, there is no observable difference in Ca^{2+} dynamics of mPFC excitatory neurons between the control group, bare MN, and Au-MN groups, which indicates the MNs have no significant damage to the brain tissue and introduce little influence on the neuronal activity.”

On pages 26-27 in *Supplementary Information* section, we added **SI Figures 21** and **22**.

On pages 2-3 in *Supplementary Information* section, we added **SI Note 2**, “**Immunohistochemical analysis** (additional)

Brain sections were then incubated for ~16 h at 4 °C in a blocking buffer containing goat anti-GFAP of Santa Cruz Biotechnology (1:1000), rabbit anti-Iba1 of Fujifilm Wako (1:500), and Chichen anti-Neuron Specific Enolase (NSE Millipore AB9698, 1:100).

All data values were presented as the mean \pm standard deviation (SD). For analyzing fluorescence density and mean of peak value in the region of interest (ROI), we define the ROI as the areas within 200 μ m lateral to the edge of the probes. One-way ANOVA was used to compare the fluorescence density and mean of peak value between different experimental groups.

As is well known, GFAP and IBA-1, representing astrocytes and microglia, respectively, reflect the inflammation in the brain tissues^{1 2}. Neuron-specific enolase (NSE), an acidic protease unique to neurons and neuroendocrine cells and a sensitive indicator for assessing the severity of nerve cell damage and prognosis, is widely used to reflect neural injuries and pathological processing³.

Stereotaxic surgery (additional)

For in vivo photometry recordings, mice were unilaterally injected with 250 nl of AAV5-CaMKII-G-CaMP6f mixed with 5 nl AAV5-CaMKII-tdTomato into the mPFC at these coordinates AP: -1.5 mm, ML: +0.3 mm, DV: -2.1 mm. Optical fibers (Newdoon Inc, O.D.:

1.25 mm, core: 200 μ m, NA: 0.37) were implanted 0.2 mm above the mPFC (AP: -1.5 mm, ML: +1.4 mm, DV: -1.9 mm). At the same time, MN probes were implanted ipsilaterally into the medial prefrontal cortex at 30 degrees lateral to the optical fiber.

Fiber photometry system

The multi-fiber photometry system was used as previously described^{4 5}. Briefly, the system consisted of a 488 nm excitation laser, a fluorescence cube, and a spectrometer. The 488 nm laser beams first launched into the fluorescence cube and then into the optical fibers. The GCaMP and tdTomato emission fluorescence collected from the fiber probe traveled back to the spectrometer. Only animals with strong GCaMP and tdTomato expression were included in the study (**SI Figure 22a**). Spectral data was acquired by OceanView software (Ocean Optics, Inc) at 10 Hz and was synchronized to a 20 Hz video recording system to acquire the animal behavior.

The in vivo recordings were carried out in an open-top home cage (21.6 " 17.8 " 12.7 cm) in the 30 Lux red light environment. Laser power was adjusted to a final optical fiber output of 30 mW. Photometry data were exported to MATLAB R2014b for analysis. Coefficients of GCaMP6f and tdTomato were unmixed by a customized script by fitting spectrum signals to standard emission curves. GCaMP6f signals were normalized by tdTomato signals for motion correction. (**SI Figure 22a**). A 0.1 Hz high-pass filter corrected the fluorescence bleaching. Photometry signals ($\Delta F/F$) were derived by calculating $(F-F_0)/F_0$, where F_0 is the median of the fluorescence signal (**SI Figure 22b**). For the home cage analysis, we recorded data for 5 min per mouse and calculated the $\Delta F/F$ to further analyze the correlation of SuM and DG signals in raw data (**SI Figure 22c**). Cumulative activity = $\Delta F/F \times \text{Time}$. The average of the peak of $\Delta F/F$ above 2SD during 5 mins in the home cage was calculated."

We added the following references in the *Supplementary Information* section, corresponding to **SI Note 2**:

1. Giovannoni, F. & Quintana, F. J. The Role of Astrocytes in CNS Inflammation. *Trends Immunol.* **41**, 805–819 (2020).
2. Muzio, L., Viotti, A. & Martino, G. Microglia in Neuroinflammation and Neurodegeneration. *Front. Neurosci.* **15**, 742065 (2021).
3. Thelin, E. P. *et al.* Utility of neuron-specific enolase in traumatic brain injury; relations to S100B levels, outcome, and extracranial injury severity. *Crit. Care* **20**, 285 (2016).
4. Meng, C. *et al.* Spectrally Resolved Fiber Photometry for Multi-component Analysis of Brain Circuits. *Neuron* **98**, 707-717.e4 (2018).
5. Li, Y. *et al.* Supramammillary nucleus synchronizes with dentate gyrus to regulate spatial memory retrieval through glutamate release. *Elife* **9**, e53129 (2020).

Comment 6: "Is the Au coated PLGA MNs completely degraded and absorbed after

administration without removal? How long does it take for PLGA microneedles to degrade in mice?”

Our response: We thank the referee for this comment. In our SOP, the implanted drug delivery interface consists of gold and PLGA. As shown in this manuscript, the degradation of gold layer is completed within a minute upon an electrical trigger. However, though being a bioresorbable polymer, the degradation of PLGA is much slower than gold. Generally, the degradation time scale of such MN varies from several weeks to several months, depending on the molecular weight, monomer ratio, thickness, and structure.¹ The implanted PLGA MN can be partially or fully resorbed by the organism given long enough time. Here, we conducted an accelerated degradation experiment of PLGA MN (3 mm, without drug payload) by soaking it in 65 °C phosphate-buffered saline (PBS) and observed changes in the shape and morphology during the immersion. It is clear that, after 24-hour degradation, the main body of PLGA MN collapsed. This experiment simulates the chronic degradation of PLGA in a biofluidic environment under ambient conditions, showing significant degradation of PLGA MN.

SI Figure 23. PLGA MN degradation in PBS. The PLGA MN (3 mm) is soaked in PBS at 65 °C as an accelerated degradation experiment. Optical images by microscope are taken before degradation (a), 30 minutes after degradation (b), 1 hour after degradation (c), 6 hours after degradation (d), and 24 hours after degradation. Scale bar: 1 mm.

Modifications to the manuscript: Based on the reviewer's comments, we have made the following revisions:

On page 16 in the revised manuscript, we added "Besides, we study the bioresorbability of the MN. Though the degradation of PLGA is much slower than gold⁷⁵, we show this process of PLGA in an accelerated degradation experiment in 65 °C phosphate-buffered saline (PBS), as illustrated in **SI Figure 23**."

On page 28 in *Supplementary Information* section, we added **SI Figure 23**.

On page 3 in *Supplementary Information* section, we added **SI Note 3**, "We conducted an accelerated degradation experiment of PLGA MN (3 mm, without drug payload) by soaking it in 65 °C phosphate-buffered saline (PBS). We observed changes in the shape and morphology during the immersion (**SI Figure 23**). The MN was taken out of the solution for observation after 30 min, 1 h, 6 h, and 24 h. It is clear that, after 24-hour degradation, the main body of PLGA MN collapsed. This experiment simulates the chronic degradation of PLGA in the biofluidic environment under ambient conditions, showing significant degradation of PLGA MN."

We added the following references in the revised manuscript:

1. Hirenkumar, M. & Steven, S. Poly Lactic-co-Glycolic Acid (PLGA) as Biodegradable Controlled Drug Delivery Carrier. *Polymers (Basel)*. **3**, 1377–1397 (2011).

Comment 7: "There are some mistakes in figure description. For example, SI Figure 12 shows the representative confocal images of 40- μ m horizontal cortical slices at various stages after implantation of the bioresorbable electrode probes, not "performance and impedance analyses of this wireless SOP". The authors should revise the manuscript in general to correct such errors. In addition, the authors also should check the references for formatting inconsistencies."

Our response: We thank the referee for this comment. We have modified the manuscript in different aspects, including figures, grammar, and references.

Regarding figure descriptions, we reorganized most of the figures are located in *Supplementary Information*. The corresponding numbers of them are updated in captions, main text, *Supplementary Information*, etc. Main **Figure 3 & 5** are also rearranged carefully, with captions and main text updated correspondingly.

Regarding references, we go through each of them to investigate possible errors. Some author names are corrected (Reference No. 2, 5, 48, 79, 80, 83), as the full surname of them was not provided in a proper way.

Some page numbers or article numbers are added (Reference No. 1, 6, 11, 13, 16, 17, 20, 22, 24, 31, 32, 33, 35, 38, 39, 43, 44, 45, 47, 51, 52, 63, 66, 68, 72, 76, 78, 88, 90). Some references are corrected in format (Reference No. 30, 42, 54, 84).

In addition, we replaced reference No. 73 with the following one, as it is more specified to the data we mentioned.

Regarding grammar and expression, we read through the entire manuscript and corrected the errors we found. For example, most of the special terms are added with “the” in the front. The single and plural form and time is also revised.

Modifications to the manuscript: Based on the reviewer’s comments, we have made the following revision:

We corrected the references in the revised manuscript, here are some examples.

2. Teo, B. M. & Seah, B. C.-Q. Recent advances in ultrasound-based transdermal drug delivery. *Int. J. Nanomedicine* **13**, 7749–7763 (2018).

17. Hsu, W. L. *et al.* On-skin glucose-biosensing and on-demand insulin-zinc hexamers delivery using microneedles for syringe-free diabetes management. *Chem. Eng. J.* **398**, 125536 (2020).

24. Li, W. *et al.* Long-acting reversible contraception by effervescent microneedle patch. *Sci. Adv.* **5**, eaaw8145 (2019).

30. Yin, Y. *et al.* Separable Microneedle Patch to Protect and Deliver DNA Nanovaccines against COVID-19. *ACS Nano* **15**, 14347–14359 (2021).

54. Tait, W. S. Electrochemical corrosion basics. in *Handbook of Environmental Degradation Of Materials: Third Edition* 97–115 (2018).

73. Neal, N. C. & Burke, F. D. High-pressure injection injuries. *Injury* **22**, 467–470 (1991).

We unify the expression of term. For example, “gold”, except for some special cases like “Au-MN”, are unified instead of using “Au”.

We check the abbreviations and make sure they are at least mentioned in the full term when they first appear. For example, “MN” for microneedle.

We check the singular or plural form of nouns and verbs. They are now consistent with what they’re used to describe. For example, to describe a microneedle array, “microneedles” and “MNs” are used instead of “microneedle” and “MN”.

We make sure that we use present tense and active voice in most cases in the main text. For example, “**SI Figure 5** provides a schematic illustration of the two-electrode system of SOP (**SI Figure 5a**) and the atomic scale illustration of the anodic oxidation (**SI Figure 5b-c**) on Au (111) facet in a neutral environment, with reaction products, Au₂O₃ in the form of nanoparticles.”

We reorganize the expressions in the manuscript to avoid redundancy and improve conciseness. For example, we use “drug delivery” instead of “the delivery of drug”.

REVIEWERS' COMMENTS:

Reviewer #1 (Remarks to the Author):

The paper has been well improved. All of my concerns are addressed with additional discussions concerning several obscures provided. Some details are also included in the revised manuscript. I am therefore pleased to see it published as its current form.

Reviewer #2 (Remarks to the Author):

The manuscript has been greatly improved, but some minor concerns should be addressed before acceptance.

1. SI Figure 13f-h lack of figure legends. Besides, please add the error bars for SI Figure 13c-h and the variance and standard curve equation for SI Figure 13c.
2. In supporting information (line 191), the authors described that the release study is carried out in 45 °C 1X DPBS. Obviously, this is inconsistent with the conditions described in the main manuscript (40 °C, line 329). Please unify the release condition truthfully. And why not choose 37°C (near the temperature of human body)?
3. What is the molecular weight of PLGA used in this work? Please add more PLGA information in the manuscript.

Responses to comments of Referee #1

Comments from Referee #1:

Summary Comment: “The paper has been well improved. All of my concerns are addressed with additional discussions concerning several obscures provided. Some details are also included in the revised manuscript. I am therefore pleased to see it published as its current form.”

Our response: We thank the referee for these positive comments and the helpful suggestions for revision. Your advice during the revision greatly helped us improve the manuscript.

Modification to the manuscript: None.

Responses to comments of Referee #2

Comments from Referee #2:

Summary Comment: “The manuscript has been greatly improved, but some minor concerns should be addressed before acceptance.”

Our response: We thank the referee for the positive comments and the helpful suggestions for revision. We carefully addressed the issues, as listed below, and revised our manuscript accordingly.

Modifications to the manuscript: None.

Comment 1: “SI Figure 13f-h lack of figure legends. Besides, please add the error bars for SI Figure 13c-h and the variance and standard curve equation for SI Figure 13c.”

Our response: We thank the referee for this comment. We apologize for missing the figure legends of **SI Figure 13f-h** during last revision. They are updated. In SI Figure 13c, we added 95% confidence band and 95% prediction band together with the linear regression line. The equation, R^2 , and the variance of coefficients are provided. The error of peak absorption was obtained by calculating different absorptions based on the horizontal (wavelength) shifting of peak. In **SI Figure 13f-h**, error bars were added to denote the prediction intervals.

SI Figure 13. Characterization of dye release from microneedles (0.3% Rhodamine B loaded, 1.2-mm). **a.** Optical images of an MN array undergoing dye release from 0 to 60 minutes in 45 °C 1X DPBS. Scale bar: 500 μm . **b.** The UV-Vis spectroscopy (300-800 nm) of Rhodamine B standard solutions (Sample 1-6). **c.** The calibration curve of Rhodamine B from standard solutions in **b**. The calibration curve is calculated as: $\text{Abs} = 0.09518 * C - 0.02998$, where Abs is the absorbance and C is the concentration. The standard error of the slope is $7.16908 * 10^{-4}$, and the

standard error of the intercept is 0.00373. The adjusted R^2 is 0.99972. The 95% confidence band and 95% prediction band are plotted together with the linear regression line. Data are presented as mean values \pm standard deviation of the peak absorption (Abs) based on the horizontal shifting analysis of peak. (n=6) **d.** Measured UV-Vis spectra of the environment solution corresponding to **a.** **e.** The absorbance versus time of environment solution from the dye release experiment. **f.-h.** The concentration, molar amount, and release ratio of Rhodamine B calculated based on calibration curve in **c** and the absorption data in **e**. Data are presented as predicted values \pm margin of error from 95% prediction interval. (n=6)

Modifications to the manuscript: Based on the reviewer's comments, we have made the following revisions:

On pages 15-16 in *Supplementary Information* section, we revised the caption of **SI Figure 13** accordingly.

Comment 2: "In supporting information (line 191), the authors described that the release study is carried out in 45 °C 1X DPBS. Obviously, this is inconsistent with the conditions described in the main manuscript (40 °C, line 329). Please unify the release condition truthfully. And why not choose 37 °C (near the temperature of human body)?"

Our response: We thank the referee for this comment. We apologize for this inconsistency. After checking the experimental notebook, we confirmed that the study was carried out in 45 °C 1X DPBS. The reason why we chose a higher temperature than 37 °C is mainly for an accelerated drug release. Meanwhile, 45 °C is not too high to break the structure of the MNs or lead to significant chemical changes.

Modifications to the manuscript: Based on the reviewer's comments, we have made the following revisions:

On pages 9, 18-20 in the revised manuscript, we corrected the temperature from "40 °C" to "45 °C".

Comment 3: "What is the molecular weight of PLGA used in this work? Please add more PLGA information in the manuscript."

Our response: We thank the referee for this comment. The Poly(D, L-lactide-co-glycolide) (PLGA) we used in this work has a molecular weight (Mw) ranging from 50000 to 75000, which is ester terminated. We added this information in the manuscript.

Modifications to the manuscript: Based on the reviewer's comments, we have made the following revisions:

On page 4 in the revised manuscript, we added "Mw = 50~75 kDa, ester terminated" in the description of PLGA.